# Economic valuation of temperature-related mortality attributed to urban heat islands in European cities

Wan Ting Katty Huang [1,2], Pierre Masselot [3], Elie Bou-Zeid [4], Simone Fatichi [5], Athanasios Paschalis [6], Ting Sun [7], Antonio Gasparrini [3,8,9] & Gabriele Manoli [1,10]

As the climate warms, increasing heat-related health risks are expected, and can be exacerbated by the urban heat island (UHI) effect. UHIs can also offer protection against cold weather, but a clear quantification of their impacts on human health across diverse cities and seasons is still being explored. Here we provide a 500 m resolution assessment of mortality risks associated with UHIs for 85 European cities in 2015-2017. Acute impacts are found during heat extremes, with a 45% median increase in mortality risk associated with UHI, compared to a 7% decrease during cold extremes. However, protracted cold seasons result in greater integrated protective effects. On average, UHI-induced heat-/cold-related mortality is associated with economic impacts of €192/€ − 314 per adult urban inhabitant per year in Europe, comparable to air pollution and transit costs. These findings urge strategies aimed at designing healthier cities to consider the seasonality of UHI impacts, and to account for social costs, their controlling factors, and intra-urban variability.

The majority of Europe's population lives in cities and suburban areas, and while the overall European population is projected to decline in the future, the population in most large cities is expected to increase[1], further increasing the share of urban dwellers. Globally, much more urban growths are expected particularly in Africa and South Asia[2]. As more people live in highly urbanized settings, greater exposure to extreme heat has been observed[3], further amplifying the impact of warming due to climate change[4–6]. This is due to the urban heat island (UHI) effect, whereby urban environments are found to be warmer than their rural surroundings, e.g.,[7,8]. Urban environments are characterized by an abundance of built structures, intense human activities, and often a scarcity of vegetation. The geometry and materials of the built structures alter the urban energy balance through reduced

evaporative cooling, increased radiative absorption, and enhanced heat storage, e.g.,[7,9–11]. Combined with additional heat emissions from anthropogenic activities, this results in the UHI phenomenon, which varies in magnitude by time and location due to the variability in meteorological, surface, and anthropogenic conditions[8,11].

Increased temperatures can have adverse impacts on human health[12,13], as illustrated by the 2003 European heatwave when 70,000 excess deaths were observed across Europe[14]. A large number of those deaths occurred in Europe's largest cities, such as Paris and Berlin[15,16]. In England, ~2000 excess deaths were estimated, with the greatest impact observed in London[17]. More generally, exposures to both heat and cold temperatures have adverse effects on human health and are associated with increased cardiovascular, respiratory, and

[1]Department of Civil, Environmental and Geomatic Engineering, University College London, London, UK. [2]Met Office, Exeter, UK. [3]Department of Public Health, Environments and Society, London School of Hygiene & Tropical Medicine, London, UK. [4]Department of Civil and Environmental Engineering, Princeton University, Princeton, USA. [5]Department of Civil & Environmental Engineering, National University of Singapore, Singapore, Singapore. [6]Department of Civil & Environmental Engineering, Imperial College London, London, UK. [7]Institute for Risk and Disaster Reduction, University College London, London, UK. [8]Centre for Statistical Methodology, London School of Hygiene & Tropical Medicine, London, UK. [9]Centre on Climate Change and Planetary Health, London School of Hygiene & Tropical Medicine, London, UK. [10]Laboratory of Urban and Environmental Systems, School of Architecture, Civil and Environmental Engineering, Ecole Polytechnique Fédérale de Lausanne (EPFL), Lausanne, Switzerland. ✉e-mail: gabriele.manoli@epfl.ch

cerebrovascular mortality risks[12,18–20]. UHIs are known to exacerbate such heat-related risks, e.g., ref. 21, but there is also evidence of protection against cold extremes[22,23], a subject that is relatively understudied. The trade-off between summer risks and winter benefits of UHI-related mortality remains largely unexplored, especially across different cities and background climatic conditions. Furthermore, the climate benefits of UHI mitigation strategies are known to vary across seasons and geographic regions, for e.g., refs. 8,24, which must be considered in urban planning and building designs, e.g., refs. 25,26. Most of the existing literature on UHI mitigation focuses on thermal comfort rather than quantifiable health risks[27,28], while studies investigating temperature-related mortality tend to focus on a few selected cities or countries[21,23,29–31] without the necessary granularity to account for intra-urban variations in temperature and population distribution.

Additionally, a previous study has shown that UHIs can double the economic impacts of global climate change[32] and, for example, it was estimated that UHI cost the city of Melbourne $300 million per year due to impacts on health, infrastructures, and energy demand[33]. The inclusion of economic assessments of mortality has also been shown to greatly increase the social cost of carbon for integrated assessment models[34]. Hence, we deem it useful to frame the debate in monetary terms in order to influence future planning and decision-making—which are largely shaped by socioeconomic rules[35] and require robust scientific knowledge to balance multiple social costs and benefits. While such economic assessments are beneficial in quantifying the human cost in cost-benefit analyses, it is important to acknowledge that these are individual lives whose value cannot truly be expressed in monetary terms. Instead, stated preference surveys, on which mortality valuations are commonly based, represent people's willingness to pay to reduce mortality risk, as is consistent with its use in cost-benefit analyses. Additionally, it should be noted that risks are generally not experienced equally across the population and can disproportionally affect more disadvantaged populations.

In this study, we provide a consistent analysis of UHI's impact on mortality risk for 85 cities across Europe and consider the impact of both heat and cold temperatures over the entire annual cycle. Mortality risks are explored across urbanization gradients using high-resolution urban climate simulations, population density data, and city-level temperature-mortality relationships. We further monetize climate-related risks and compare the economic valuation of UHIs' mortality impact with other urban living costs.

## Results

Following the Intergovernmental Panel on Climate Change (IPCC) convention, risk in this study is defined as "the potential for adverse consequences"[36], mainly in the form of human mortality. IPCC's definition of exposure is "the presence of people…in places and settings that could be adversely affected"[36]. However, except where otherwise specified, risks in this study are assessed from an individual's perspective, for each city's average inhabitant. The same average inhabitant is always assumed to be exposed, and intra-city risk differentials are indicative of the potential difference in risk posed to this inhabitant if they lived in one part of the city compared to another. Exposure in this study therefore follows the epidemiological definition and refers to the presence of heat and cold. Under IPCC convention, this is more closely related to the presence of hazard ("the potential occurrence of a natural or human-induced physical event or trend that may cause loss of life…"[36]). Lastly, vulnerability is "the propensity or predisposition to be adversely affected…, including sensitivity or susceptibility to harm and lack of capacity to cope and adapt"[36]. Vulnerability determines the magnitude of impact of a given temperature exposure. In the current context, vulnerability may include, among others, physiological factors such as age and acclimatization to a particular climate, as well as societal and socioeconomic factors that hinder people's ability to cope

with heat or cold, such as access to air conditioning. In this study, vulnerability and exposure, jointly, determine risk.

### Temperature-related mortality risk in European cities

Mortality risk generally increases towards the extreme ends of each city's typical temperature range, with greater risks to older age groups (Fig. 1a–c). The degree of sensitivity to heat and cold differs between cities due to differences in population vulnerability, associated with local socioeconomic, infrastructural, and environmental characteristics[37,38]. The intensity of UHI, defined as the difference in average air temperature between urban areas and their rural surroundings, also varies between cities and across seasons (Fig. 1a–f, Supplementary Fig. S7)[8,24,39]. In this study, the differences in UHI intensity between cities are captured by the urban climate simulations through variations in the land surface, local climate, and human activities, while seasonality results from seasonal shifts in both climate (e.g., solar radiation) and anthropogenic heat emissions (e.g., central heating)[40]. The combined effects of varying both vulnerabilities and UHI (exposure) lead to variations in UHI impact on mortality risk across Europe. For instance, focusing on the heat extreme days, a greater impact is noted in Milan, which experiences more extreme heat conditions, than in London, despite the greater urban-rural temperature difference in the latter city. This is due to the strong non-linear increase in heat-related risk under Milan's heat extreme days, which are notably warmer than in London (Fig. 1). Stockholm, on the other hand, has a relatively weak UHI and low vulnerability to heat, resulting in minimal impact of UHI on mortality risk.

On an annual basis, UHIs lead to a general shift in the temperature distribution to warmer conditions in urban environments (Supplementary Fig. S8). This results in greater mortality risk associated with heat and lower risk associated with cold compared to rural surroundings. The annual net impact, therefore, further depends on the balance of warm and cold days per year, in addition to each city's vulnerability to heat vs. cold.

### Impact of urban heat islands on mortality risk

Despite the binary classification into urban and rural environments, a range of built and natural environments can be found within each. Examined as a function of land imperviousness, we find the largest change in mortality risk per unit change of land imperviousness around the urban-rural boundary (slopes in Fig. 2a–c), from rural conditions of minimal imperviousness (Δ imperviousness = 0%) to urban environments with slightly built-up areas (e.g., Δ imperviousness = 5%). This is followed by a continued but more gradual increase/decrease within the urban imperviousness gradient. The general shape of this relationship largely reflects that between imperviousness and air temperature UHI (Supplementary Fig. S12)[9,41]. The degree of increase/decrease in risk with imperviousness is further associated with each city's UHI intensity and population vulnerability, as discussed in the next section.

Given that highly populated areas also tend to be highly built-up (statistically significant mean Spearman's correlation of 0.47, Supplementary section 1), and that heat emissions from human activities are enhanced in densely populated areas, UHI's impact on mortality risk also increases with population density (Supplementary Fig. S1). At the city level, this implies a biased exposure whereby greater fractions of the population live in areas of greater UHI. Given that risks in this study are assessed from the perspective of an individual inhabitant living in different parts of the city, this bias in exposure is not captured. Should it be included, however, the bias would result in a 6.4% median (interquartile range across 85 cities, IQR: 2.9–9.4%) increase in the urban average mortality risk estimate during heat days and a 1.3% (IQR: 0.7–2.3%) reduction during cold days (Supplementary section 1). For the outlier cases of Brussels, Dublin, Paris, and Lyon, the increase during heat days can reach over 25%.

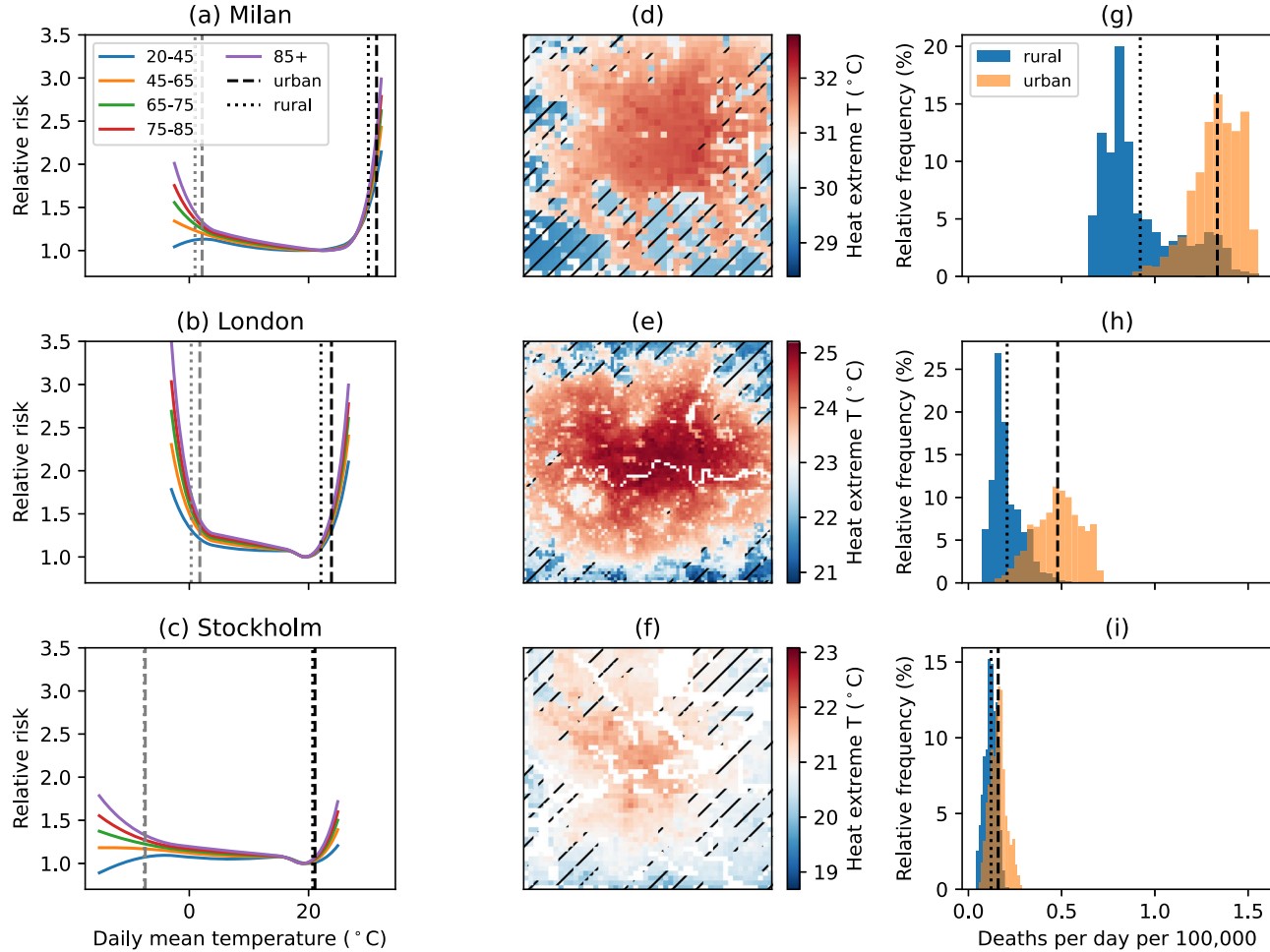

**Fig. 1 | Example cities. a–c** Age grouped exposure-response relationships, **d–f** temperature averaged over the warmest 2% days between 2015 and 2017, and **g–i** histograms of average temperature-related mortality estimates during the warmest 2% days shown separately for urban and rural areas for **a, d, g** Milan, **b, e, h** London, and **c, f, i** Stockholm. Vertical dashed lines in black/gray in **a–c, g–i** are the urban average over the warmest/coldest 2% days; vertical dotted lines, are the rural average. Hatching indicates rural areas in **d–f**, where areas excluded from analysis (water bodies and areas with elevation differential >100 m from the population-weighted average) are shown in white.

Grouping cities by the Köppen–Geiger climate classification (Supplementary Table S3) and summing the impact over all days of the year, we note that temperate cities with no dry season and hot summers tend to experience an annual net adverse effect of UHI on mortality, while for all other climate groups, the net effect is generally protective (Fig. 2a). However, variability and outlier cities can be found within each climate group, as well as overlaps between climate groups (Supplementary Fig. S9). During days of extreme heat, the impact of UHI on mortality risk tends to be stronger for cities with temperate, hot summer climates (Fig. 2b), though this difference is removed and the impacts become similar for all climate groups during heat extremes when population age is standardized between cities (Supplementary Fig. S5b). On the other hand, consistently and notably weaker UHI protective effects are found for cold climate cities during cold extremes days, regardless of assumed population age structure (Fig. 2c, Supplementary Fig. S5c). The differences between climate groups are further discussed in Supplementary section 3.

In general, UHIs have a weak protective net annual impact on human mortality risk for most (70 out of 85; 90% confidence interval, CI: 60 to 75) cities examined in this study (Fig. 3a, b). Across all 85 cities, a median of 2.8 fewer deaths per 100,000 people per year (CI of the median estimate, representing uncertainty in exposure-response relationships: 1.7–3.5; IQR across 85 cities, representing differences between cities: 0.7–4.4) are attributed to

temperature in urban versus rural areas. This is due to the high likelihood of cold days in most parts of Europe, defined as cooler than the city- and age-specific optimal temperature below which mortality risk generally decreases with increasing temperature. The most protective annual net effects of UHI are found in Glasgow, Porto, and London, with 16 (CI: 14–18), 12 (CI: 11–14), and 11 (CI: 9–13) fewer deaths per year per 100,000 people, respectively (Supplementary table S4). The most adverse annual net effects are found in the urban areas of Trieste, Genoa, and Bologna, with 5 (CI: 2–7), 4 (CI: 1–8), and 3 (CI: 1–5) additional temperature-related deaths per year per 100,000 inhabitants, respectively (Supplementary Table S4). On average, UHI has a protective effect in all cities for all seasons except summer (Fig. 3a). In some cities with cooler summer climates, such as those in Nordic countries and the United Kingdom, the average impact is protective even in summer.

The daily impact of UHI on mortality risk is greater during heat extreme days than during cold extreme days for all cities. Over the 85 European cities examined, UHI increases mortality risk during heat extreme days by a median of 0.25 (CI: 0.21–0.27; IQR: 0.18–0.29) additional deaths per 100,000 population per day (Fig. 3a, c), a median increase of 45% (IQR: 30–61%) from the risk in rural areas around the cities (Supplementary Fig. S10). This is in contrast to a reduction of 0.05 (CI: 0.04–0.07; IQR: 0.02–0.11) deaths per 100,000 population per day during cold extremes (Fig. 3a, d), a 7% (IQR: 3–15%) decrease

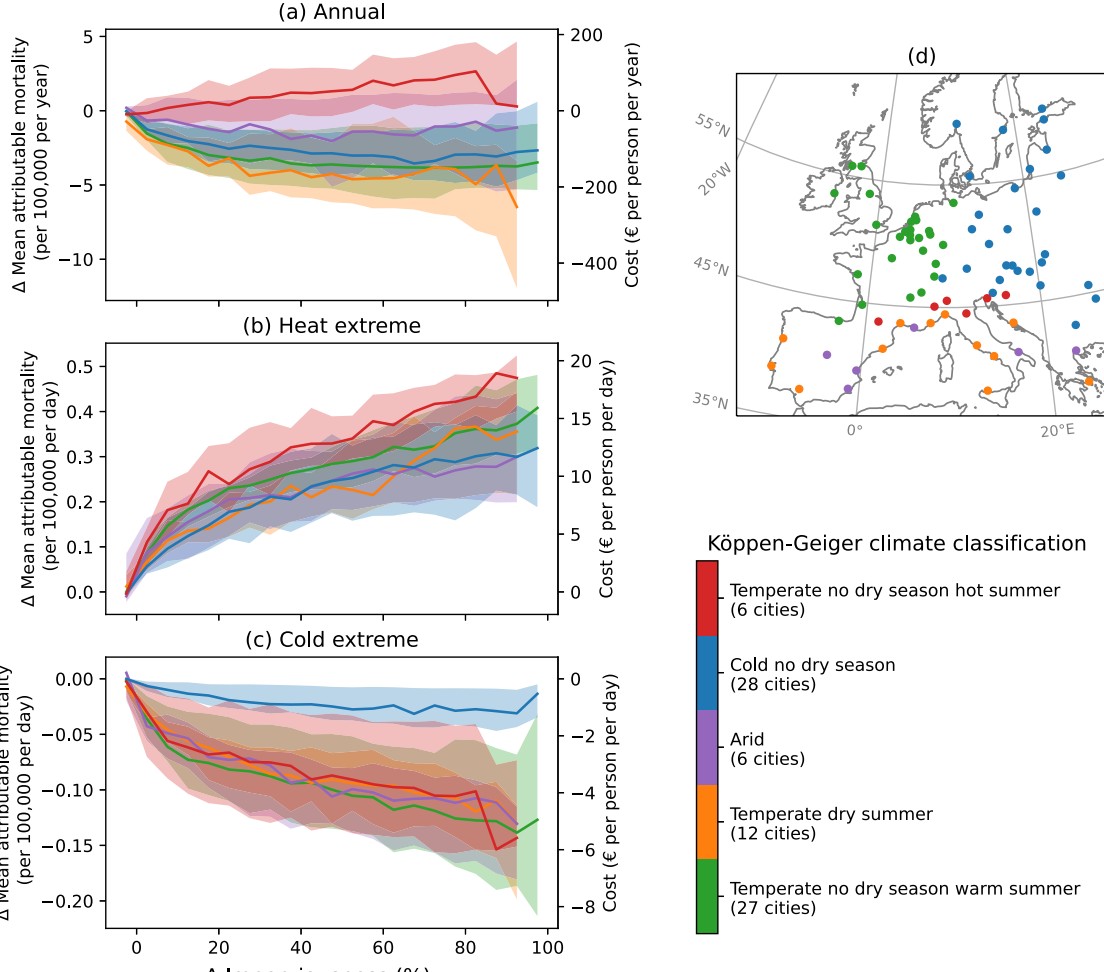

**Fig. 2 | Difference in attributable mortality, and associated economic impact, compared to the rural mean, as a function of the difference in land imperviousness from the rural mean.** The annual net impact per year across all days of the year is shown in **a**, followed by the average per day over the **b** warmest and **c** coldest 2% (22) days over the 2015–2017 time period. Cities are grouped by the Köppen-Geiger climate classification, as indicated in **d** and listed in Supplementary Table S3. Solid lines in **a**–**c** indicate the climate group median, with shading showing the interquartile range. Cities where the maximum urban-rural imperviousness difference is less than 80% are excluded (67 of 85 cities shown), and medians across less than four cities in a climate group, which can occur at high Δ imperviousness, are not shown to avoid misrepresentation.

from the rural average. Within the urban space, the impact is further enhanced in more built-up parts of the city. For instance, there is a 67% (IQR: 46–91%; 0.35, IQR: 0.24–0.44, additional deaths per 100,000 per day) median risk differential between the most and least built-up (≥90% vs. ≤10% imperviousness) areas during heat extreme days (Supplementary Fig. S11).

If population age structures are standardized across the cities according to the 2013 European standard population, the risk estimates would increase by 9% on median (Supplementary section 2, Fig. S4c), likely indicative of slightly younger populations in cities (Supplementary Fig. S4a).

To additionally consider the age dependence of UHIs' impact on mortality, years of life lost (YLL) analyses are included in the Supplementary materials (Fig. S13, Tables S6 and S10). While the overall conclusion of a weakly protective annual net impact for most cities still holds, an annual adverse impact is found for more cities with the YLL approach (20 vs. 15 with the mortality counts-based analysis). As younger age groups are weighted more strongly in YLL analyses, this finding may be reflective of younger populations' greater vulnerability to heat, which is more similar to that of older age groups, compared to their vulnerability to cold, which tends to be lower (Fig. 1a–c).

## Factors controlling UHI's impact on mortality

The magnitude of UHI impact on human mortality in this study is determined by four factors: vulnerability of the population (as captured by the relative risk, RR, at different temperatures) and population age structure, which together determine total vulnerability, and frequency of occurrence of different temperatures and magnitude of UHI, which together determine exposure. The relative importance of each in determining how the impact of UHI compares between different cities is discussed below.

Mortality risk tends to increase towards the extreme ends of a city's typical temperature range, resulting in U- or J-shaped temperature-mortality exposure-response functions (ERFs). Subsequently, an indicator of a population's vulnerability to heat and cold is the magnitude of the relative risk (RR) at the warmest and coldest extremes of the observed temperature range ($RR_{Tmax}$ and $RR_{Tmin}$, respectively). As expected given the approach we use for estimating mortality, statistically significant and notable rank correlations can be found between $RR_{Tmax}$ and UHI's impact on mortality during days of extreme heat, as well as between $RR_{Tmin}$ and UHI's impact during extreme cold (Fig. 4), indicating greater impact in more vulnerable cities. This vulnerability factor also partially explains the difference in UHI impact between cities of different Köppen–Geiger climate

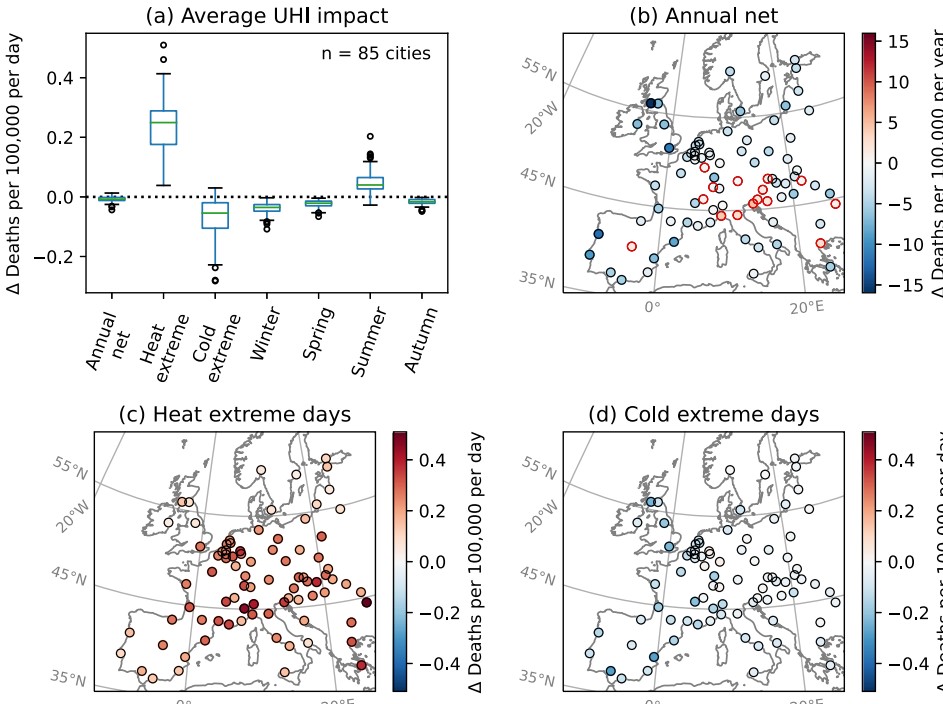

**Fig. 3 | Average urban heat island impact on temperature-related mortality risk for adult inhabitants in each city, expressed as the urban average minus the rural average.** Differences are calculated daily and then temporally averaged over the days/seasons indicated. Boxplots in **a** show the spread across 85 European cities and the variation between different temporal averages. Maps show geographical distributions of the average impact **b** annually, and over the **c** warmest and **d** coldest 2% days over the 2015–2017 period. Boxes in **a** indicate the median and the first and third quartiles, whiskers the minimum/maximum value within 1.5 times the interquartile range from the first/third quartiles, and dots the outliers. Cities, where UHI has an adverse annual net impact, are outlined in red in **b**.

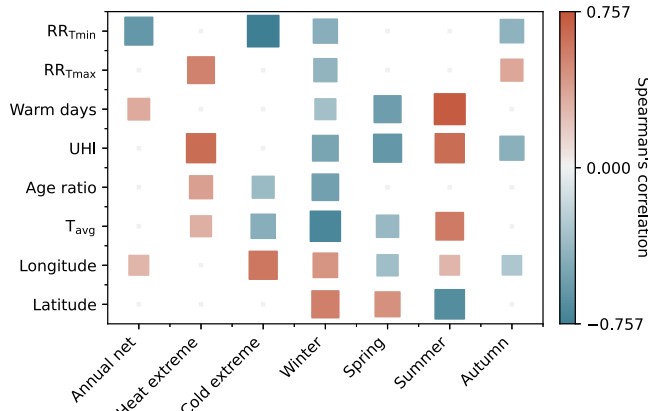

**Fig. 4 | Two-sided Spearman's rank correlation between city's mortality risk averaged across different time periods and the city's characteristics.** Metrics examined include the relative risk (RR) at the extreme ends of the city's temperature range, the number of days in a year warmer than the minimum mortality temperature for the 65–74 age group, the average magnitude of the urban heat island (UHI), the ratio of the population aged 85+ compared to those aged 20–44, the annual average temperature ($T_{avg}$), as well as the longitude and latitude. The size of the square corresponds to the magnitude of the correlation. Only correlations with statistical significance above 99% (*p* value <0.01) are shown. Negative correlations indicate a greater protective effect or lower adverse effect with a higher value of the metrics.

groups during days of temperature extremes (Supplementary section 3). Notably, the annual net impact of UHI on health is significantly correlated with $RR_{Tmin}$ but not with $RR_{Tmax}$, indicating an overall dominant role of impact during cold days across European cities.

Not all summer days are considered warm (above the Minimum Mortality Temperature, MMT, above which UHI has an adverse effect on human health) for all cities (Supplementary Fig. S15), and during mild summer days, UHI has minimal or slightly protective effect on health. The number of warm days in a year therefore strongly determines UHI's impact in summer, and to a lesser extent in spring, winter, and on the annual net balance (Fig. 4). The number of warm days in a year is correlated with the average temperature and anti-correlated with the latitude, but varies between cities independently of their population's vulnerability to heat (Supplementary Fig. S14).

The magnitude of each city's average UHI is correlated with the magnitude of its impact on health both during heat extreme days and on seasonal averages (Fig. 4). However, the magnitude of UHI is not a key factor in determining the relative ranking of each city's annual net and cold extreme mortality impact. In those cases, the population's vulnerability to cold plays the most important role.

Lastly, the proportion of older compared to younger adult populations in each city is a determining factor in the impact ranking between cities only during temperature extreme days and in winter (Fig. 4), though it affects the magnitude of the impact in each city, as discussed above and in Supplementary section 2.

Geographically, cities in eastern and northern parts of Europe, as well as those with colder average temperatures, tend to experience less protective effects of UHI on mortality during winter and cold days (Figs. 4, 3d, and 5b), possibly indicative of greater infrastructural and behavioral adaptation to cold weather. During summer, on the other hand, cities with warmer climates and in the southern and eastern parts of Europe tend to experience a stronger overall adverse impact, likely associated with a greater proportion of warm days. The daily impact during heat extremes, which is more correlated with population vulnerability and magnitude of UHI, does not follow this geographical pattern (Figs. 4 and 3c).

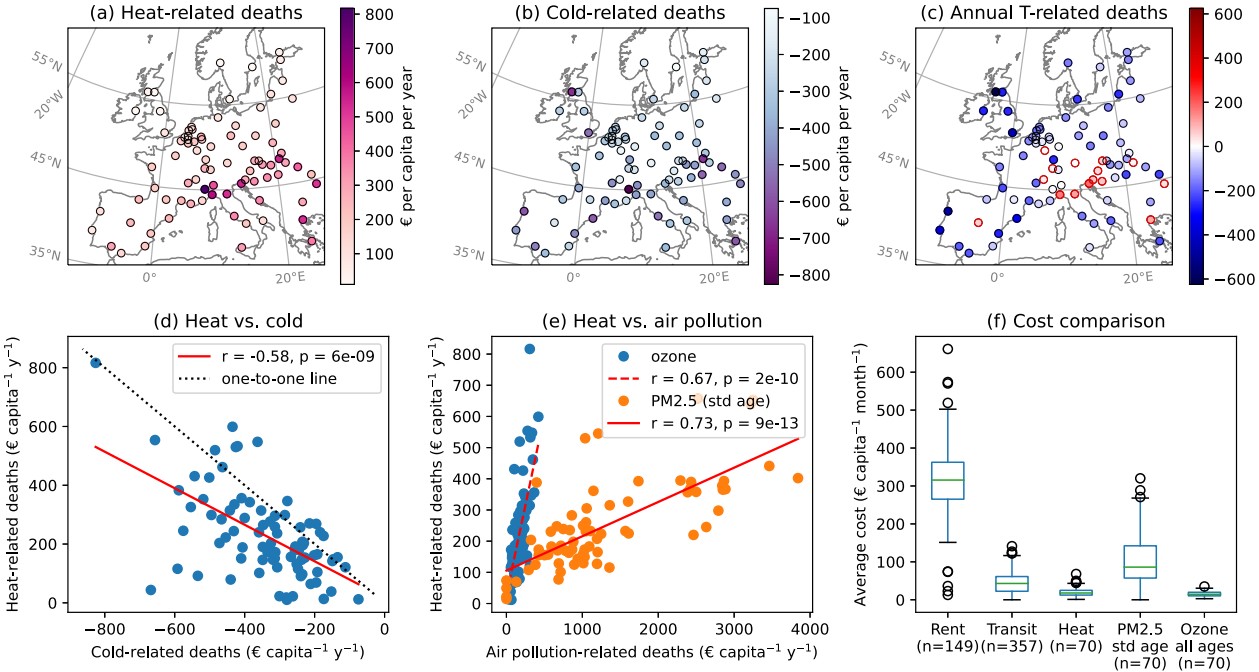

**Fig. 5 | Economic assessments of mortality risk in 2021 Euros per adult city inhabitant per year.** Annual impact of UHI-induced mortality associated with **a** heat, **b** cold, and **c** all temperatures. **d** shows the two-sided correlation between annual heat and cold UHI impacts. **e** shows two-sided correlations between heat-related mortality and air pollution-related mortality impacts for a subset of 70 cities. **f** shows comparisons of the economic impacts of mortality to costs of rent and public transport as obtained from Eurostat. The n-number in brackets indicates the number of cities represented by each boxplot. Note that due to limited data availability, rent and transport costs are representative of different subsets of European cities than heat- and air pollution-mortality and may be biased toward some countries (e.g., the majority of rent data is for German cities). Economic assessments of mortality are age-standardized according to the 2013 European standard population where indicated (std age). Only adult city inhabitants are accounted for all impacts except for ozone, which considers the entire population of all ages. Cities, where UHI has an adverse annual net economic impact, are outlined in red in **c**. Pearson correlation coefficients and associated *p* values are labeled in **d**, **e**. Boxes in **f** indicate the median and the first and third quartiles, whiskers the minimum/maximum value within 1.5 times the interquartile range from the first/third quartiles, and dots the outliers.

## Economic assessment of UHI-related mortality

Economic assessments in this study mainly follow the value of statistical life (VSL) approach, which is commonly used in mortality risk valuations[42]. However, other valuation metrics may also be applied, and the value of life year (VOLY) approach is additionally assessed and discussed below at the end of this section.

Economic assessments of UHI heat- and cold-related mortalities in Europe following the VSL approach are on the order of a few tens to hundreds of Euros per adult city resident per year (Fig. 5a, b). The median economic impact of UHI heat-related mortality is €192 (IQR: €142 to 296) per adult resident per year, estimated in 2021 Euros, and that of cold-related mortality is €−314 (IQR: €−429 to −235) per adult resident per year. There is a weak correlation between the magnitudes of cities' heat- and cold-related mortality impacts (Fig. 5d). The correlation persists even when population age is standardized (Supplementary Fig. S16d), though many cities represent exceptions. For instance, UHI yields considerable protection against cold in Glasgow, London, and Porto, while the annual heat-related mortality impact due to UHI remains low (lower left corner in Fig. 5d, Supplementary table S4). UHI-related mortality has an adverse net economic impact for 15 (CI: 10 to 25) of the 85 cities examined (circled in red in Fig. 5c, above the one-to-one line in Fig. 5d, Supplementary Table S4), with the greatest impacts for Turin and Bologna.

To put these figures in context, we compare these economic assessments with those for existing estimates of excess mortality due to air pollution[43,44], which is another important human health hazard associated with urban environments. Across Europe, cities with higher economic impacts of UHI heat-related mortality tend to also bear greater impacts from air pollution-related mortality (Fig. 5e). In contrast, the correlation is weaker between UHIs' protective effects against cold and each city's air pollution-related risk (Supplementary Fig. S16e). On an annual basis, the median economic impact of UHI heat-related mortality is around one-fifth that of PM$_{2.5}$-related mortality and ~1.2 times that of ozone-related mortality (Fig. 5f). However, it is important to note that UHI heat risk is a quantification of the urban-rural difference, while for the air pollution risks discussed above, city average pollutant concentrations are contrasted against an ideal threshold concentration that may also be exceeded in rural and peri-urban regions. This is especially the case for ozone, where higher concentrations can be observed in rural regions downwind of urban centers, e.g., ref. 45 and urban neighborhoods with lower imperviousness and more abundant vegetation[46]. Additionally, heat risk has a strong seasonality and is largely absent during colder seasons. Air pollution, on the other hand, is present throughout the year despite some seasonal variability. Moreover, mortality quantifications for air pollution used in this comparison[43,44] are based on annual average estimates, while temperature-related risks are based on daily average estimates. The former, therefore, mainly represents the impact of chronic exposure while the latter captures acute effects (with lag periods of up to three weeks, see Methods). The variability in economic impacts of UHI heat-related mortality across Europe is relatively small compared to that of PM$_{2.5}$ mortality (Fig. 5f).

Economic assessments by VSL as presented above do not explicitly consider age, therefore life expectancy at the time of death is assumed to be comparable to all causes of death. However, as can be observed from age-dependent temperature-mortality relationships (Fig. 1a–c), heat and cold exposure disproportionately affect older populations. The life expectancy at the time of death would therefore likely be shorter than the average for all-cause mortality. An approach to account for this is to consider the YLL through VOLY valuation,

which is included in Supplementary Table S7. While there is a strong correlation between economic impacts as determined through VSL and that through VOLY (Supplementary Fig. S17a, b), the magnitude of the impact as assessed through VOLY is only ~14% (median of the multi-city median annual net, heat, and cold impacts) of that assessed through VSL (Supplementary Fig. S17c). This is mainly due to differences in valuation approaches which resulted in each VSL (3.91 million 2021-EUR per statistical life[42]) equating to 85 VOLY (46,000 2021-EUR per year[47]). However, there is currently no clear consensus on economic assessments by YLL, and at the higher end of the VOLY estimate (116,000 2021-EUR per year[47]), each VSL equates to around 34 VOLY, resulting in valuations by VOLY-based assessments to be around 35% of that using VSL (Supplementary Fig. S17d). Given limitations with the VOLY approach, including ethical concerns and lack of evidence in the assumption of a time-independent VOLY[42], VSL-based assessments are the main focus of this study.

## Discussion

UHI is found to have the greatest impact on mortality risk during heat extreme days, with a median of 0.25 (CI: 0.21–0.27; IQR: 0.18–0.29) additional deaths per 100,000 adult city inhabitants per day, or 45% (IQR: 30–61%) increase in comparison to rural areas, across 85 European cities over each city's warmest 2% (22) days in 2015–2017. This is in contrast to a lower protective effect during cold extreme days, with a median decrease of 0.05 (CI: 0.04–0.07; IQR: 0.02–0.11) deaths per 100,000 adult city inhabitant per day, or 7% less (IQR: 3–15%) compared to rural areas. Despite this, the median annual balance of adverse and protective impacts of UHI on temperature-related mortality is minimal, due to the cold-to-mild climate during most of the year for most cities in Europe. This indicates the need for UHI mitigation strategies[48] that account for the seasonality of risk, similarly to the accounting for the seasonality of UHI's impact on energy consumption[49,50]. As cold- and warm-season UHIs are not necessarily interlinked, there is potential to reduce the heat-related risk without sacrificing the cold-related benefits. For example, evapotranspiration cooling of urban vegetation is mainly effective during summertime[24,51]. A modeling study of reflective cool roofs as a summer UHI mitigation strategy in the UK has also found negligible impact on winter mortality risks[52], though this is dependent on the regional climate and the type of mitigation strategy considered[23,53]. Innovations in urban materials can also introduce solutions that are optimized for each season, such as thermochromic roof membranes, the albedo of which changes with season[54]. Additionally, climate change is expected to lead to increases in the proportion of warm days, and therefore heat-related risks, relative to cold days and cold-related risks in future[55]. Given the weak annual net protective effect of UHI for most cities examined and the greater impact per day for heat, temperature increases in the future can quickly shift the balance and lead to an annual net adverse impact. It should also be noted that 2015, 2016, and 2017 are relatively average years for European temperatures, so the impact of exceptional heatwave or cold spell events, such as the one being experienced at the time of writing of this study in 2023, would be considerably greater than the estimates for temperature extreme days provided here. Lastly, the impact of cold weather tends to be spread out over a longer period after exposure while that of heat tends to be concentrated on or shortly after the exposure day[56]. Thus, despite similar cumulative impacts on mortality, heat and cold weather have different implications for healthcare and services.

In monetary terms, the estimated median annual net economic impact of UHI heat-related mortality is €191 (IQR: €142–296) per adult city inhabitant per year following the VSL approach. However, UHI can also have other economic impacts, such as those associated with morbidity (e.g., healthcare, reduced quality of life)[57], productivity (e.g., for people who work outdoors or indoors in the absence of air conditioning)[58], and energy usage (e.g., air conditioning)[31].

Infrastructure failures during heatwaves may also be exacerbated by UHI effects[59]. Additionally, heat affects cognitive abilities and has been found to inhibit learning[60], which would impose economic impacts on longer timescales. While no UHI-specific valuation is available, a recent study on the economic impact of heatwaves in Europe found a loss of 0.3-0.5% of European GDP per year associated with loss of productivity during recent hot years[58]. In southern Europe, this can even exceed 1% of the GDP per year. In another economic assessment of air pollution's impacts on health, morbidity is estimated to account for around 30% of the total health cost[61]. Therefore, while the economic assessment of mortality presented here can provide a first-order valuation of UHI, the true total economic impact of UHI is likely to be considerably greater. This highlights the need for further analysis, especially under future climate scenarios. Analyzing the economic impacts of UHIs is becoming increasingly critical since these are the true costs of the no-action scenario when cities do not attempt to mitigate extreme urban heat. Such analyses can then allow a more even comparison of the costs of various UHI mitigation plans.

Additionally, it is important to note that temperature-related mortality is quantified in this study based on city-level exposure-response relationships, which capture the average vulnerability of the city population separated by age groups. This allows for the isolation of risk differentials across the city due to differences in outdoor temperature. However, between urban and rural communities, potential differences in speed of intervention, access to healthcare, and population health, may also have an impact on health outcomes that are not represented by city-level exposure-response relationships. Additional spatial heterogeneity in other vulnerability factors, such as properties controlling the temperature of the housing stock associated with both structural features (e.g., insulation, number of floors) and internal environmental regulation (e.g., ventilation, air conditioning), may compound the impact associated with outdoor UHI.

A previous study has found, for instance, that within Parisian neighborhoods experiencing the greatest heat exposure, those who are the most socioeconomically deprived have a twofold higher mortality risk compared to the least deprived[62]. Analysis for the current study using the 2019 English Index of Multiple Deprivation (IMD)[63] for the cities of London and Leeds, for example, also shows statistically significant, though moderate correlations between socioeconomic deprivation and UHI-induced mortality risk during heat extreme days (Spearman's correlations of −0.4 and −0.6, respectively, with deprivation decreasing with increasing IMD rank, Supplementary Fig. S18a, c). Similar, though inverted, correlations can be noted during cold extreme days (Sprearman's correlations of 0.4 and 0.6, respectively, Supplementary Fig. S18b, d). More socioeconomically deprived neighborhoods tend to experience greater UHI impact on average, either protective or adverse, though similar UHI impacts are also observed in neighborhoods with notably different deprivation levels (Supplementary Fig. S18). In addition to indicating a biased exposure for more socioeconomically deprived neighborhoods, this also highlights a potential overlap between neighborhoods with higher UHI and those with greater vulnerability. The 2019 English Index of Multiple Deprivation[63] includes many factors that may increase the vulnerability of populations in deprived areas such as lower incomes, lower health scores, lower living standard metrics, and increased barriers to housing and social services. The greater vulnerability on top of higher exposure may therefore further compound the overall risk and impact of UHI, either positive or negative.

Moreover, to capture the true exposure experienced by the population, indoor/outdoor temperatures, as well as the mobility of the population throughout the day, would need to be properly considered, especially as there is a greater tendency for people to travel to highly built-up city centers for work[64], and thus toward higher UHIs. Additionally, each year, around 60% of the European population participates in tourism away from their place of residence for at least one

night (data code: tour_dem_totot[65]). With peak tourist season in the third quarter of the year (data code: tour_dem_toq[65]) when heat extremes also tend to occur, a seasonal factor can further mediate the impact of mobility on population exposure.

Other factors not explicitly considered in this study include the specific role of high nighttime temperature in mortality risk. This may be especially relevant for heat in urban areas as air temperature UHI is more pronounced during nighttime[9], and high nighttime temperatures have been noted to be additionally hazardous to health during heat spells[66,67]. Exacerbating effects of long heatwaves and cold spells, on top of the short-term lagged response to initial exposure (which is included here), as well as the exact time of year when extreme events occur (e.g., early season vs. late season), are also not considered.

From an economic perspective, the health impacts of UHI quantified in the present study, along with that associated with ozone and PM$_{2.5}$ air pollution in cities, are found to be comparable in order of magnitude to the transport costs borne by city dwellers (Fig. 5f), which, together with rent costs, are key factors influencing the planning and long term evolution of cities[68,69]. Hence, given the urgency of shaping cities for health[70], we highlight the importance of health and social costs (as considered here, but also other ones) being included in future urban planning and decision-making processes.

For future studies on UHI health risks, greater granularity in the social vulnerabilities, individual indoor/outdoor exposures, and population mobility should be considered, as data become available, in order to capture spatial and temporal heterogeneities in exposure and risk within the urban domain. The granularity has already allowed for more focused local policy strategies against air pollution[71] and should be equally considered for heat and cold effects. This will also allow for better quantification of the role of social inequality in heat and cold exposure[72], thus paving the way to identifying strategies for healthier and more equitable cities.

## Methods
### Data
Hourly near-surface air temperature (2 m above ground level) of 100 European cities are obtained from the 100-m simulations by UrbClim—an urban boundary layer climate model[40]—via the Copernicus Climate Change Service[73]. UrbClim contains a land surface scheme coupled to a 3-D atmospheric boundary layer module. The surface energy balance is modeled for three land cover types in each model grid (urban, vegetated, and bare soil, considering a total of 15 different surface classes), which are represented using a bulk approach without detailed subgrid features. Model boundary conditions include hourly synoptic meteorology from ERA5 reanalysis (relating to wind, humidity, atmospheric pressure, radiation, precipitation, temperature, and soil moisture), as well as terrain and city properties for each grid cell. Anthropogenic heat flux is prescribed as a sensible heat flux to surface air, based on a 0.0416-degree resolution dataset from the National Center for Atmospheric Research (NCAR) Climate and Global Dynamics Laboratory (CGD)[74]. Air is assumed to be homogeneously mixed within the urban canyon, with air temperature at roof height representative of the canyon as a whole. UrbClim has been validated against observations at various cities and has been found to perform similarly to other, including more complex, models[40,75–77]. For computational considerations, daily averaged temperatures re-gridded to 500 m resolution for the years 2015 to 2017 are used in this study.

For analysis of UHIs, urban/rural and land/sea surface type classifications are also obtained from the same Copernicus dataset[73]. The classification for each model grid is defined according to the 2012 CORINE land cover class, which considers 44 classes based on satellite images and in situ data[78]. Urban areas are defined as continuous regions of urban structures and transport networks with greater than 80% impermeable land cover. Commercial and industrial sites as well as ports are excluded even if the impermeability criterion is met, while

urban areas with more than 20% scattered greenery are also excluded. When re-gridding to 500 m resolution, the dominant urban/rural land cover type is considered for each coarse grid, while for land/sea masking, only grids containing no water features are classified as land.

Additional data used in the analysis include an elevation map from Multi-Error-Removed Improved-Terrain Digital Elevation Model (MERIT DEM) at 3 arc-second resolution[79], 2015 population density data from NASA Socioeconomic Data and Applications Center (SEDAC) at 30 arc-second resolution[80], 2015 land imperviousness density data at 100-m resolution from Copernicus Land Monitoring Service[81], Köppen-Geiger climate classification maps at 1 km resolution[82], as well as the following data from Eurostat[65] - Nomenclature of Territorial Units for Statistics (NUTS) 3 level: annual mortality by age and sex (data code: demo_r_magec3, last accessed 2021-07-23), population by age and sex at the start of the year (data code: demo_r_pjangrp3, last accessed 2021-03-11), and annual population structures by age (data code: demo_r_pjanind3, last accessed 2021-03-11). Life expectancy by age and sex at NUTS 2 level (data code: demo_r_mlifexp, last accessed 2021-06-03) from Eurostat[65], averaged across the years 2015 to 2017, is used for quantifying the YLL.

Rent for a selection of cities is obtained from Eurostat[65] where available, and is calculated by multiplying the average area of living accommodation in m$^2$ per person (data code: urb_clivcon SA1022V, last accessed 2022-08-19) by the average annual rent for housing per m$^2$ (data code: urb_clivcon SA1049V, last accessed 2022-08-19). Public transport cost is represented by the cost of a monthly public transport ticket within 5-10 km of the city center (data code: urb_ctran TT1080V, last accessed 2022-08-19[65]). Rent and public transport costs in the last ten years (2012-2021) are considered and averaged where available.

Age standardized mortality for the adult population (20 years and older) associated with annual mean PM$_{2.5}$ concentrations above the World Health Organization (WHO) recommended level of 10μg/m$^3$ in European cities in 2015 is obtained from Khomenko et al.[43]. Annual premature deaths associated with ozone exposure, averaged over the years 2015 to 2017 and for the whole population of all ages and without age standardization, is obtained from the European Environment Agency[44,83]. The 2021 version of WHO's concentration-response function for ozone[84] is assumed, with a counter-factual concentration of 70μg/m$^3$.

In the discussion of results, tourism data by country and aggregated to European levels from Eurostat (data codes: tour_dem_totot and tour_dem_toq, last accessed 2023-05-23[65]), as well as the 2019 English Index of Multiple Deprivation from the UK government[63] are also quoted.

### Epidemiology
Temperature-mortality exposure-response relationships for five age groups (20–44, 45–64, 65–74, 75–84, and 85+) in each city are obtained from, and explained in more detail in, Masselot et al.[37], which follows a two-stage framework. In the first stage, quasi-Poisson regression is applied using distributed lag non-linear models (DLNM[85]) to model the relationship for a subset of cities in country-dependent age groups. Daily mortality time series from the multi-country multi-city (MCC) collaborative research network and 9 km resolution daily mean temperature from ERA5-Land are used. Lagged effects up to three weeks after exposure are considered in the DLNM and summed to represent the short-term cumulative impact of each exposure day. In the second stage, model coefficients describing the lag-cumulative exposure-response relationships are modeled in a meta-regression with age and city characteristic terms describing each city's environmental and socioeconomic states. Residual spatial differences not explained by city characteristics are captured using ordinary kriging. The exposure-response model coefficients, describing the relationship in relative temperature percentiles, for all cities and age groups are subsequently predicted. The R codes for this

analysis are available as provided in Masselot et al.[37]. Subsequently, in the current work, the models are translated to exposure-response relationships as functions of city mean temperatures using each city's UrbClim domain average temperature from the 2008 to 2017 period, where simulated microclimate is available. These relationships quantify the mean risk for each city- and age-specific population for each daily mean temperature, irrespective of factors such as UHI that led to the observed temperature. They are assumed to identically apply to the whole population in each age group in each city, irrespective of differences in each individual's actual exposure or vulnerability, i.e., it is the mean estimate. Uncertainties in the exposure-response relationships as represented by 1000 Monte Carlo simulations[37] are used to quantify the confidence intervals of the attributable mortality for each city. These simulations are then used to quantify the confidence interval of median estimates and the estimated number of cities experiencing adverse annual net UHI impacts.

### Attribution

With the above-described temperature data and exposure-response relationships, the daily mortality fraction attributable to temperature exposure is determined using the R function attrdl, modified for cumulative exposure-response relationships. This is equivalent to

$$AF = \frac{RR - 1}{RR} \qquad (1)$$

where AF is the attributable fraction and RR is the relative risk[86]. Daily attributable numbers are calculated by multiplying the attributable fraction by the per population annual average all-cause mortality for each corresponding NUTS 3 region and age group, averaged over the years 2015 to 2017. Annual values are used due to a lack of finer-scale age-specific data.

### Urban heat island impact analysis

The impact of UHI on mortality risk is computed for each day and city by taking the difference between the average attributable mortality number over urban areas and the average over rural areas within the model domain. Model grids containing water bodies and those with average elevation 100-m higher or lower than the domain population density-weighted average elevation are excluded from analysis. When computing the total impact across age groups, the NUTS 3 level local population age structure is used. Additional analysis with population age-standardized across all cities according to the 2013 European standard population[87] is also provided for ease of comparison between cities and for comparison to other health impact assessments.

Within the temperature range observed for each city, the age group-dependent minimum mortality temperature (MMT) is defined as the temperature with the lowest cumulative associated mortality risk, according to each age group's temperature-mortality ERF. Each day is then subsequently classified as either warm or cold for each age group, depending on whether it is warmer or cooler than the MMT. Mortality risk generally increases both with increasing temperature above the MMT and with decreasing temperature below the MMT. The age dependence of MMT, and therefore the age dependence of whether a condition is considered warm or cold, implies that the same temperature may be associated with heat-related mortality for one age group and cold-related mortality for another. Heat and cold extremes, on the other hand, are the warmest/coldest 2 % (22) days (as defined by the UrbClim domain population-weighted daily average temperature) over the three-year period.

When classifying cities into Köppen-Geiger climate groups, the dominant climate classification within the UrbClim urban domain for each city is considered. Given that some cities belong in climate groups not otherwise represented by our sample of 85 cities,

climate groups with only one city are merged with the closest matching group and labeled by their common base classification. For instance, Murcia with "Arid, desert, and hot climate" is grouped with six other cities with "Arid, steppe, and cold climate" into the Arid climate group.

### Economic assessment

Economic impacts of mortality are quantified following the VSL approach. The best estimate of 3.6 million 2005-USD per life lost for EU-27 countries is obtained from the Organization for Economic Co-operation and Development (OECD) report on Mortality Risk Valuation in Environment, Health and Transport Policies[42]. This is derived from quality-screened samples in a meta-analysis of stated preference surveys[88].

To translate this into relevant values for Europe, the VSL is first converted to 2005-EUR using the 2005 Euro area (19 countries) purchasing power parities (PPP). Inflation is then accounted for by considering the ratio of the Euro area consumer price index (CPI) in 2021 compared to that in 2005. i.e.,

$$VSL_{2005-USD} \times PPP_{2005,\,Euro} \times \frac{CPI_{2021,\,Euro}}{CPI_{2005,\,Euro}} = VSL_{2021-EUR} \qquad (2)$$

where PPP and CPI values are for the 19-country Euro area, obtained from OECD[89,90]. $VSL_{2005\text{-}USD}$ is 3.6 million 2005-USD, $PPP_{2005,\,Euro}$ is 0.853 EUR per USD, $CPI_{2021,\,Euro}$ is 107.8, $CPI_{2005,\,Euro}$ is 84.7, and $VSL_{2021\text{-}EUR}$, used for valuation in this study, is 3.91 million 2021-EUR.

To compare with economic impacts associated with air pollution, $PM_{2.5}$- and ozone-related mortality from refs. 43 and 83 are translated into monetary terms following the same VSL valuation. Comparisons are made only for the cities where the risks associated with all exposures are available (74 cities).

Economic assessments of mortality can also additionally account for the age at the time of death by quantifying by YLL. This is calculated by multiplying the number of premature mortality in each age group by the average life expectancy in that age range. The value of the life year is estimated to be 40,000 2010-EUR per life year[47], translated into 46,000 2021-EUR with a $CPI_{2010,\,Euro}$ of 93.1[89]. Note, however, that there is high uncertainty in this estimate, with bounding estimates[47] of 29,000–116,000 2021-EUR.

### Reporting summary

Further information on research design is available in the Nature Portfolio Reporting Summary linked to this article.

## Data availability

Data generated from the current study have been deposited on Zenodo and can be accessed at https://doi.org/10.5281/zenodo.7986841. Data used in the current study are described in detail in the Data section under Methodology above. UrbClim data can be obtained from Copernicus Climate Change Service (https://doi.org/10.24381/cds.c6459d3a), elevation map from MERIT DEM (http://hydro.iis.u-tokyo.ac.jp/yamadai/MERIT_DEM), population density from NASA SEDAC (https://doi.org/10.7927/H49C6VHW), land imperviousness data from Copernicus Land Monitoring Service (https://land.copernicus.eu/pan-european/high-resolution-layers/imperviousness), Köppen-Geiger climate classification from https://www.gloh2o.org/koppen/, Eurostat data from https://ec.europa.eu/eurostat, data on mortality associated with $PM_{2.5}$ from the supplementary materials associated with Khomenko et al.[43] (https://doi.org/10.1016/S2542-5196(20)30272-2), data on mortality associated with ozone from the European Environment Agency (https://www.eea.europa.eu/data-and-maps/data/air-quality-health-risk-assessments, permalink: https://www.eea.europa.eu/ds_resolveuid/86ef37b3bf844299b978867c86cf99e7), and temperature-mortality relationships from https://doi.org/10.5281/zenodo.7672108.

## Code availability

Mortality attribution analyses in the current study were performed in R version 4.0.5. Subsequent data analyses were performed in Python version 3.9. Core code associated with the analyses is available on https://github.com/hkatty/Paper_UHI_mortality_Europe[91].

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

## Acknowledgements

W.T.K.H. and G.M. acknowledge the support of The Branco Weiss Fellowship—Society in Science administered by ETH Zurich. In later stages, W.T.K.H. is further supported by the Met Office Hadley Center Climate Program funded by DSIT. E.B.Z. is supported by the Army Research Office under contract W911NF2010216. S.F. acknowledges the support of the National University of Singapore through the project 'Bridging scales from below: the role of heterogeneities in the global water and carbon budgets', Award No. 22-3637-A0001. A.P. is supported by the Natural Environment Research Council NERC-funded CAMELLIA project (Community Water Management for a Liveable London) in London (NE/S003495/1). T.S. is supported by UKRI-NERC Independent Research Fellowship (NE/P018637/2). A.G. is supported by the Medical Research Council-UK (Grant ID: MR/V034162/1) and the European Union's Horizon 2020 Project Exhaustion (Grant ID: 820655).

## Author contributions

G.M. conceived and supervised the project. W.T.K.H. devised and performed the analyses and drafted the manuscript. P.M. and A.G. developed the ERFs and provided guidance on their use and interpretation. E.B.Z., S.F., A.P., and T.S. provided input on the interpretation of the results. All co-authors provided feedback that helped shape the analysis and the manuscript.

## Competing interests

The authors declare no competing interests.
