## [Peer Review File · Nature Communications]

Economic valuation of temperature-related mortality attributed to urban heat islands in European citiesREVIEWER COMMENTS

Reviewer #1 (Remarks to the Author):

Dear Authors,

This are my comments:

Key results

This article presents a complex study on the effects of the UHI on public health, evaluating its potential economic impact. Results are very consistent and adequately address the complexity of the Topic. This article accurately addresses the singularities of climate contexts and the contrasting effect of UHI during the warm and cold seasons. Most articles tend to only focus on the negative effects of UHI, unlike this article which identifies both its positive and negative impacts, providing an economical valuation for diverse climate contexts, that should be seen as a referential for the assessment of the UHI.

Validity

The authors use a large dataset, although its data were obtained by modeling. Despite its limitations, which are adequately addressed, results should be seen as reliable, since the Model has been previously validated.

Significance

Results are very significant since they adequately expand the current knowledge by accurately addressing the complexity of this topic, demonstrating that some of its most common assumptions, such as the negative impact of UHI, deserve a more complex interpretation such as the one presented in this article. Results can be used for urban planning, and to address climate adaptation strategies.

Data and methodology

This article uses modeled data that seems consistent with the reality that is addressed. Data is adequately presented and graphics are, in most cases, complete and easy to read.

However, the Authors are asked to revise the following elements:

- Figure 1 – Please provide the units of measurement for the y-axis density.
- Some of the images of the supplementary materials lack the necessary quality to be adequately understood, such as the cases of figure S8 and Figure S15.
- Tables S2 and S3, should repeat the heading lines on each page to improve their legibility.

Methodologies are adequately presented and seem appropriate.

On lines 393-394, you mention "Residual spatial differences not explained by city characteristics are captured using ordinary kriging." For the benefit of replicability, please mention which software tool was used for this purpose.

On lines 431-434 - Please explain the phrase: "Sub-classifications with only one city are merged with the closest matching classification into one climate group (e.g. Murcia with Arid, desert, and hot climate is grouped with six other cities with Arid, steppe, and cold climate into the Arid climate group)."

Analytical approach

The analytical approach is in my opinion correct as the methods used for addressing the UHI intensity

are clear and adjusted. Statistical analysis is also adjusted to the objectives of the study. I don't feel qualified to fully judge the quality of the methods behind epidemiology and cost estimation, however, they are clear and seem appropriate.

Suggested improvements

In my opinion, this article requires little improvement as it already provides a multilayer approach to a complex phenomenon. Therefore, I will only suggest some minor changes:

On Lines 62 and 63, the authors mention: "g. 8, 24], but there are no quantitative guidelines on how urban planning can deliver potential health benefits.". This is a rather controversial argument, since several articles and Books offer quantitative proposals for urban interventions, presenting case studies or modeling potential changes, here are some examples:

- <https://doi.org/10.1016/j.esd.2022.05.006>

- Erell, E., Pearlmutter, D., & Williamson, T. (2012). Urban microclimate: designing the spaces between buildings. Routledge.

Lines 170-172 – "Compared to when the standard population age structure is assumed, considering the local age structures yields a median doubling of UHI's impact estimate on all mortality risk metrics" I – Not quite sure what the authors meant to say.

Clarity and context

This article provides clear and accessible language. Most of its results are very clear and should be easily understood by both experts and the general public.

References

References are adjusted and provide an adequate literature background to this article's contents. However, it seems out of format in the present document

Your expertise

My expertise is in UHI analysis and urban climate planning. Aspects related to Epidemiology and Cost estimates would be evaluated in more depth by reviewers from those areas, however, from my knowledge it seems like a quality article.

For the overstated reasons I am suggesting that this article should be published after minor revision.

Reviewer #2 (Remarks to the Author):

This work studies the impacts of temperature changes due to urban heat island (UHI) on mortality, across 85 European cities belonging to 5 background climatic conditions. It accounts for intra-urban variations in temperature and population distribution and compute the health and economic impacts over the entire annual cycle, as well as the net effect.

The conclusions are original by providing results on the net balance of the benefits of UHI over the complete year, for many cities covering various climatic conditions. They can help the decision-maker grasp the impact of UHI on short term mortality, and consider policies aimed at reducing them at the city level. The conclusions and data interpretation are robust, valid and reliable, with many supplementary analyses that support the main results (see however my comments below). The references seem adequate to me.

I have suggestions of clarifications / additional work (see below Data and methodology (I) and

suggested improvements below (III)).

I) Data & methodology

Data and methodology seem up to date, of high quality and clearly presented. However, I have some comments regarding the reporting of some results and methodological aspects.

I-1) Reporting

a) Lines 112-14: In the sentence "from rural conditions of minimal imperviousness (Δ imperviousness = zero) to urban environments with slightly built up areas (e.g. Δ imperviousness = 5)", shouldn't " Δ imperviousness = 5" instead be " Δ imperviousness = 5%" to be consistent with Figures 2(a)-(c)?

b) Lines 173-77: Regarding the sentence "The median annual UHI impact during heat days decreases from 15.0 additional deaths per 100,000 population per year in urban areas to 6.8 additional deaths when using the standard population age structure (Fig. 3a,b). The median annual protective effect during cold days also decreases from 18.9 to 9.1 fewer deaths per 100,000 population per year." I don't understand where we can see this from figures 3 (a)-(b). Indeed, these figures represent the mean per day whereas the text provides the median per year. Furthermore, the text mentions "heat days" and "cold days", which refers to extreme heat / cold days or to summer / winter seasons? Do these comments really apply to these figures? I wonder, because the next sentence in the text describes other results in these figures properly: "The highest daily UHI impact on mortality risk decreases from 1.05 additional deaths per 100,000 population per day (Budapest, Hungary; Fig. 3a) to 0.61 additional deaths per day (Bucharest, Romania; Fig. 3b), both during heat extremes".

c) Line 181: Does figure 3d really show the impact on mortality risk estimation? I have the impression that the word "this" applies rather to the 85+:20-44 population ratio for the city vs. for the standard population.

I-2) Methodology

My main expertise is in economics, although I have been working with epidemiologists for over 25 years. This may explain why the epidemiological methodology did not seem to me to be detailed enough in the text or in section 4 Methods, subsection 4.2 Epidemiology.

A) Epidemiological methodology issues

It essentially refers to an article of which Pierre Masselot (one of the co-authors of the present work) is also the author: Lines 384-86 "Temperature-mortality exposure-response relationships for five age groups (20-44, 45-64, 65-74, 75-84, and 85+) in each city are obtained from, and explained in more detail in, Masselot et al. [34]". Yet this publication is neither correctly cited in the bibliography ([34] Masselot, P. et al. Excess mortality attributed to heat and cold in 801 cities in Europe), nor a priori accessible on the net.

This led me to three questions of understanding.

a) Lines 72-4: "Mortality risks are explored across urbanisation gradients using high resolution urban climate simulations, population density data, and city level temperature mortality relationships". The authors use population-specific relationships for each city, which provide a good level of precision. However, in doing so, these relationships already incorporate differences due to UHI in the calculation of mortality risks (see Figure S8), before applying them to temperature variations related to UHI and calculating the frequency weighted UHI impact by temperature. What is the influence of these temperature-mortality relationships on the computations?

b) Lines 253-54: "The former therefore includes the impact of chronic exposures while the latter only capture acute effects (with lag periods of up to three weeks, see Methods)." A lag of 3 weeks is

mentioned, referring to the Methods section, but this section does not give any further details. What are the implications of this “up to three weeks” lag in the calculations, particularly those associated with extreme heat and cold days? Are the impacts on mortality considered to be spread over three weeks? What happens if several consecutive days are classified as extreme? Is there a harvesting effect?

c) Lines 317-21: It is suggested to “capture the true exposure experienced by the population”. In addition to the points rightly raised by the authors, more generally, deaths seem to be associated with people's place of residence. If people move around the country or abroad during the hottest 2% of days (often in the summer, during holidays), how are deaths counted if they do not occur in their place of residence? What proportion of people are in their main residence during heatwaves?

B) Other methodological issues

a) The place of residence differs from the place of work (generally in UHI), which the authors acknowledge. This involves (on average) a higher exposure to high temperature during day w.r.t the place of residence. Is there evidence in the literature of the magnitude of the discrepancies involved?

b) Are the conditions of health care (speed of intervention, transport to a health care facility) during the health event causing the death the same between UHI and non-UHI? Can this influence mortality between residents in and out of HUI?

C) Remarks on economic issues

I agree with the value of statistical life (VSL) chosen to assess mortality impacts, which was carefully calculated by the OECD in 2012 and widely used since then. I am also satisfied with the way it is translated into a value relevant for Europe. Note, however, that some countries recommend that the VSL should be changed according to changes in Gross Net Product per capita instead of Consumer Price Index, but the differences are limited. I have three points to make.

a) First, I am reluctant to the use of the terms “costs” and “benefits” when associated with non-market impacts, here the valuation of mortality. This can be misleading to the reader as these amounts will not actually be gained or lost by the cities concerned, rather they are welfare losses in their population associated with premature mortality and valued economically. I prefer the terminology “economic impacts” or “economic assessments”, which can be negative or positive. I would then be satisfied if this costs/benefits formulation was changed throughout the text. This includes the abstract “UHIs in Europe cost/save €480/€677 in heat-/cold-related mortality”, as well as the title “Mortality costs and savings of urban heat islands in European cities”. For the latter, something like “Economic valuation of acute mortality attributed to urban heat islands in European cities” would be better.

b) Second, in choosing this VSL, the economic computations assume that life expectancy at death is similar to that observed for all causes of death. Since the effects acute mortality are being assessed here, it would be useful to discuss the expected reduction in life expectancy. A difference in life expectancy, and the extreme case of a single year of life lost per death attributable to UHI temperature conditions, would alter the economic computations.

c) Related to the point above, I find that the authors too quickly eliminate the possibility of conducting an analysis based on years of life lost (lines 453-57), and a valuation that would calculate them on the basis of ages of death (at least, age groups) and exposition response functions. This would indeed take into account at least the loss of life expectancy associated with age (even if not with cause). This was done by Khomenko et al (2021) for PM2.5 for example. It would not be necessary to go as far as monetarization (even if some values of life year lost exist for Europe, such as in Desaiques et al., 2011), but at least to make a net balance of years of life gained/lost that would take into account the

age distribution.

II) Use of statistics

In the box-and-whisker plots (Figures 3a et b, as well as S3, S7, S10 S14 and S16), please specify what the values represent (certainly the median for the centre line, quartiles 1 and 3 for the box, 1.5 the interquartile range (IQR) Q3-Q1 for the whiskers (with the respective boundaries the minimum and maximum), and outliers for dots outside the IQR).

III) List of suggested improvements

Although this work is already very comprehensive, I think that the benefits of implementing some of the 4 directions I present below are worth considering.

a) The authors compare their results with those for mortality associated with PM2.5 exposure. The latter is, in general, proportional to population density in cities, and thus acts in synergy with the effects of temperature for UHI, both in summer and winter. However, another pollution indicator appears to be related to heat wave mortality: ground-level ozone (e.g. Reid et al, 2012). In this case, the relationship with UHI would be rather antagonistic in summer or even on hot spring or autumn days, since in highly polluted areas, and under certain insolation conditions, high NOx concentrations can destroy part of the ozone. Consequently, ozone levels are likely to be higher in peri-urban and rural areas downwind of UHI. To what extent can this affect the results?

b) Men and women have a different reactivity to temperature stress. This is linked to physiological factors: muscle mass, which is higher on average in men, generates heat, whereas fat mass, which is higher on average in women, does not. It is also linked to different lifestyles: fewer women are active, implying differences in exposure at work or when travelling to work. The authors seem to have the annual mortality by age and sex. If other data are available by sex too, would it be possible to perform the calculations in step 4.2 separately by sex?

c) The authors discuss the consequences of climate change on their analyses. Lines 276-81, it is written: "Additionally, climate change is expected to lead to increases in the proportion of warm days, and therefore heat-related risks, relative to cold days and cold-related risks in future [47]. Given the weak annual net protective effect of UHI for most cities examined and the greater impact per day for heat, temperature increases in future can quickly shift the balance and lead to an annual net adverse impact". Why not make these estimates? Gasparrini et al. (2017) did so on the basis of different representative concentration pathways (RCPs). Why not also apply it to UHI under climate change scenarios? It seems to me that the authors can easily obtain all the data they need. And this will eventually allow to determine, for each city, when the adverse effects of high temperature UHI will outweigh the protective effects of low temperature UHI.

d) In Line 337, the authors mention in future work a more detailed study of vulnerabilities. Why would it not be possible to make a first attempt to construct indicators of deprivation and to match them at grid level? Several indexes are possible.

- The European Deprivation Index within urban areas, based on EU – SILC (European Union – Statistics on Income and Living Conditions) can be exploited, or build from aggregated census data and tax incomes of the population by year.

- A simpler version, originally constructed in France (Windenberger et al., 2012), is based on 4 variables: the median household income per consumption unit, the percentage of high-school graduates in the population aged 15 years and older, the percentage of blue-collar workers in the active population aged 15 to 64, and the unemployment rate in the active population aged 15 to 64.

- The Townsend deprivation index (Townsend, 1987), originally constructed in England and Wales, is also based on four variables: the rate of households without a car, the rate of non-owner households, the rate of households with more than one person per room, and the rate of unemployment in the active population.

IV) References

- Desaigues, B., Ami, D., Bartczak, A., Braun-Kohlová, M., Chilton, S., Czajkowski, M., Farreras, V. et al. (2011) Economic valuation of air pollution mortality: A 9-country contingent valuation survey of value of a life year (VOLY). *Ecological Indicators* 11, 902–910
- Gasparrini, A., Guo, Y., Sera, F., Vicedo-Cabrera, A. M., Huber, V., Tong, S., ... & Armstrong, B. (2017). Projections of temperature-related excess mortality under climate change scenarios. *The Lancet Planetary Health*, 1(9), e360-e367
- Guillaume, E., Pornet, C., Dejardin, O., Launay, L., Lillini, R., Vercelli, M., Marí-Dell'Olmo, M., et al. 2016. Development of a Cross-Cultural Deprivation Index in Five European Countries. *Journal of Epidemiology and Community Health* 70(5), 493–99.
- Khomenko, S., Cirach, M., Pereira-Barboza, E., Mueller, N., Barrera-Gómez, J., Rojas-Rueda, D., ... & Nieuwenhuijsen, M. (2021). Premature mortality due to air pollution in European cities: a health impact assessment. *The Lancet Planetary Health*, 5(3), e121-e134.
- Reid, C. E., Snowden, J. M., Kontgis, C., & Tager, I. B. (2012). The role of ambient ozone in epidemiologic studies of heat-related mortality. *Environmental health perspectives*, 120(12), 1627-1630.
- Townsend, P. (1987). Deprivation. *Journal of Social Policy*, 16(2), 125-146.
- Windenberger, F., Rican, S., Jougl, E., & Rey, G. (2012). Spatiotemporal association between deprivation and mortality: trends in France during the nineties. *The European Journal of Public Health*, 22(3), 347-353.

Reviewer #3 (Remarks to the Author):

The article is innovative and brings a significant contribution to the research in the field of urban heat island and mortality under heat and cold conditions. The study area is wide-ranging, and all the most advanced methodologies have been applied to investigate the relationship between mortality and temperature. The use of UrbClim makes the analyses (partially) integrated with the characteristics of the built environment and the Köppen-Geiger climate classification ensures that the classifications used are validated and integrated with the current literature. The graphs are accurate and understandable. Finally, the inclusion of micro-particle pollution is an added value, although it would have been interesting to see the association between O3 and PM10, 2.5 and heat- and cold- related deaths.

Review of the paper:

“Mortality costs and savings of urban heat islands in European cities”

First of all, I would like to congratulate the authors for the excellent work. The conducted analyses are in line with the up-to-date scientific literature and the discussion of the results is very rich and interesting. This manuscript brings a significant contribution to the research in the field of urban heat island and mortality under heat and cold conditions within EU urban areas. The most advanced methodologies have been applied to investigate the relationship between mortality and temperature and the use of UrbClim makes the analyses (partially) integrated with the characteristics of the built environment. In addition, the Köppen-Geiger climate classification ensures that the classifications used are validated and integrated with the current literature. The inclusion of micro-particle pollution is an added value, although it would have been interesting to see the association between O₃ (in summer) and PM_{10, 2.5} (in winter). Notwithstanding, contents and technical details are of excellent quality.

As minor comment, I would like to raise two points:

1] A better clarification of the meaning of “*exposure*”, “*vulnerability*”, and “*risk*” in the context of this research would not be a useless explanation. Especially in the climate change field, many of these concepts/words blur getting sometimes meanings sensibly different according to the reader’s background. For this reason, I believe that a clear definition of the used terms would be of help to close the gaps between the environmental epidemiological and other interdisciplinary studies (see IPCC Glossary, IPCC 2022).

2] A clear identification of social inequalities in the nexus between extreme temperatures and mortality would have provided a clearer understanding of the risk identification across the different categories of the population (most educated vs least educated, most isolated vs least isolated, etc.), but we are aware that the availability of data at the urban level is not the same in all cities. Based on the results produced by the study (e.g., *Table S2 “Mortality associated with UHI, per 100,000 urban adult population”*), it might be interesting to further investigate these aspects in some cities where similar data are available.

Other punctual recommendations in the text:

On Results:

Line 95: insert a “,” after “*e.g.*”

Line 126: it could be interesting to see some examples “*of cities which reach around the 20% of enhancement*”.

On Discussion:

- The use of the term “*mitigation*” here can lead to misunderstandings. I would suggest using other terms in line with the IPCC and UNFCCC languages (i.e., “*adaptation*”) (IPCC 2022), when appropriate.

Reference in the text:

Line 267: “*this indicates the need for mitigation strategies*”

Line 299: “*such analyses can then allow a more even comparison to the costs of various mitigation plans*”

- Spatial heterogeneity could be influenced by outdoor (e.g., direction of the windows front, building age, roofing construction, number of floors) as well as indoor (e.g., thermal isolation, walls and ceiling, fans, double-glazed windows) environmental characteristics. Consider whether to quote them in the text.

On Methods:

- No citations to DLNM model in the text (section 4.2).
- No citations to MMT in the text (section 4.4).

Line 397: about “*UrbClim domain average temperature from the 2008 to 2017 period*”. Why did you consider only 9 years? Is the available timeframe? For the World Meteorological Organization, 30-year reference period reflects changing climate.

Line 313-316: In relation to “*A previous study has found, for instance, that within Parisian neighbourhoods experiencing the greatest heat exposure, those which are the most socioeconomically deprived have a twofold higher mortality risk compared to the most affluent [53]*”. To provide a clearer overview of the state-of-the-art in relation to social inequalities - extreme temperatures – health, I suggest referring to other cities when considering the relation between temperature and mortality and different health outcomes socio-economic variables. One option may be to quote studies over cities with a higher mortality ranking associated with UHI (Table S2 in Supplementary Material) (e.g., Turin, Madrid, Bologna, Thessaloniki, etc.).

Dear reviewers,

Thank you very much for your constructive feedback and suggestions which led to improvements of the manuscript. In response to your comments, we have now included additional analyses on ozone pollution, years of life lost, and socioeconomic deprivation. During the revision, we also fixed a bug in the age structure analysis. This brought the local age-based risk estimates closer to that using the standard population. All manuscript text on that topic, along with all local age-based analyses and figures, have now been revised and corrected. Please find below in indented blue text replies to individual questions and comments. The manuscript with tracked changes is attached separately. Thank you again for your time and consideration.

Best wishes,

Katty Huang, on behalf of all co-authors

Reviewer #1 (Remarks to the Author):

Dear Authors,

This are my comments:

Key results

This article presents a complex study on the effects of the UHI on public health, evaluating its potential economic impact. Results are very consistent and adequately address the complexity of the Topic. This article accurately addresses the singularities of climate contexts and the contrasting effect of UHI during the warm and cold seasons. Most articles tend to only focus on the negative effects of UHI, unlike this article which identifies both its positive and negative impacts, providing an economical valuation for diverse climate contexts, that should be seen as a referential for the assessment of the UHI.

Validity

The authors use a large dataset, although its data were obtained by modeling. Despite its limitations, which are adequately addressed, results should be seen as reliable, since the Model has been previously validated.

Significance

Results are very significant since they adequately expand the current knowledge by accurately addressing the complexity of this topic, demonstrating that some of its most common assumptions, such as the negative impact of UHI, deserve a more complex interpretation such as the one presented in this article. Results can be used for urban planning, and to address climate adaptation strategies.

Data and methodology

This article uses modeled data that seems consistent with the reality that is addressed. Data is adequately presented and graphics are, in most cases, complete and easy to read.

However, the Authors are asked to revise the following elements:

- Figure 1 – Please provide the units of measurement for the y-axis density.

Thank you for pointing this out. The values and units for density is dependent on the histogram bins (counts per bin width), which isn't very meaningful, so this is now changed to show the frequency instead, with the units properly labelled.

- Some of the images of the supplementary materials lack the necessary quality to be adequately understood, such are the cases of figure S8 and Figure S15.

Thank you for the comment. We interpret the comment regarding the quality of the figures to reflect their resolution. In the revised version all figures have been produced in a vector format with very high resolution for optimal clarity. More explanatory text has also now been added to the captions, in case this is what you meant instead.

- Tables S2 and S3, should repeat the heading lines on each page to improve their legibility.

Thank you for the suggestion. This has now been added.

Methodologies are adequately presented and seem appropriate.

On lines 393-394, you mention "Residual spatial differences not explained by city characteristics are captured using ordinary kriging." For the benefit of replicability, please mention which software tool was used for this purpose.

The R code for this analysis is publicly available as open source through Masselot et al. 2023. The reference has now been updated and a note has now been added in this section.

On lines 431-434 - Please explain the phrase: "Sub-classifications with only one city are merged with the closest matching classification into one climate group (e.g. Murcia with Arid, desert, and hot climate is grouped with six other cities with Arid, steppe, and cold climate into the Arid climate group)."

Sorry for the confusion. In some instances, a climate group would contain only one city in our sample of 85 cities, and in these cases the climate group with a singular city would be merged with the closest matching climate group. This has now been reworded a bit in the manuscript text as well, so hopefully the meaning is clearer.

Analytical approach

The analytical approach is in my opinion correct as the methods used for addressing the UHI intensity are clear and adjusted. Statistical analysis is also adjusted to the objectives of the study. I don't feel qualified to fully judge the quality of the methods behind epidemiology and cost estimation, however, they are clear and seem appropriate.

Suggested improvements

In my opinion, this article requires little improvement as it already provides a multilayer approach to a complex phenomenon. Therefore, I will only suggest some minor changes:

On Lines 62 and 63, the authors mention: “g. 8, 24], but there are no quantitative guidelines on how urban planning can deliver potential health benefits.”. This is a rather controversial argument, since several articles and Books offer quantitative proposals for urban interventions, presenting case studies or modeling potential changes, here are some examples:

- <https://doi.org/10.1016/j.esd.2022.05.006>

- Erell, E., Pearlmutter, D., & Williamson, T. (2012). Urban microclimate: designing the spaces between buildings. Routledge.

Thank you for pointing this out and for the references. Indeed, this statement was too broad and has now been removed. The two references provided have also been added to the manuscript.

Lines 170-172 – “Compared to when the standard population age structure is assumed, considering the local age structures yields a median doubling of UHI’s impact estimate on all mortality risk metrics” I – Not quite sure what the authors meant to say.

Thanks for pointing this out. We meant that by taking into consideration the local population age structure, the median UHI impact estimate across all cities is doubled for all measures. This sentence has been modified in the manuscript to hopefully bring more clarity.

Clarity and context

This article provides clear and accessible language. Most of its results are very clear and should be easily understood by both experts and the general public.

References

References are adjusted and provide an adequate literature background to this article's contents. However, it seems out of format in the present document

Your expertise

My expertise is in UHI analysis and urban climate planning. Aspects related to Epidemiology and Cost estimates would be evaluated in more depth by reviewers from those areas, however, from my knowledge it seems like a quality article.

For the overstated reasons I am suggesting that this article should be published after minor revision.

Reviewer #2 (Remarks to the Author):

This work studies the impacts of temperature changes due to urban heat island (UHI) on mortality, across 85 European cities belonging to 5 background climatic conditions. It accounts for intra-urban variations in temperature and population distribution and compute the health and economic

impacts over the entire annual cycle, as well as the net effect.

The conclusions are original by providing results on the net balance of the benefits of UHI over the complete year, for many cities covering various climatic conditions. They can help the decision-maker grasp the impact of UHI on short term mortality, and consider policies aimed at reducing them at the city level. The conclusions and data interpretation are robust, valid and reliable, with many supplementary analyses that support the main results (see however my comments below). The references seem adequate to me.

I have suggestions of clarifications / additional work (see below Data and methodology (I) and suggested improvements below (III)).

I) Data & methodology

Data and methodology seem up to date, of high quality and clearly presented. However, I have some comments regarding the reporting of some results and methodological aspects.

I-1) Reporting

a) Lines 112-14: In the sentence “from rural conditions of minimal imperviousness (Δ imperviousness = zero) to urban environments with slightly built up areas (e.g. Δ imperviousness = 5)”, shouldn’t “ Δ imperviousness = 5” instead be “ Δ imperviousness = 5%” to be consistent with Figures 2(a)-(c)?

Thank you for spotting this. This has now been corrected in the manuscript.

b) Lines 173-77: Regarding the sentence “The median annual UHI impact during heat days decreases from 15.0 additional deaths per 100,000 population per year in urban areas to 6.8 additional deaths when using the standard population age structure (Fig. 3a,b). The median annual protective effect during cold days also decreases from 18.9 to 9.1 fewer deaths per 100,000 population per year.” I don’t understand where we can see this from figures 3 (a)-(b). Indeed, these figures represent the mean per day whereas the text provides the median per year. Furthermore, the text mentions “heat days” and “cold days”, which refers to extreme heat / cold days or to summer / winter seasons? Do these comments really apply to these figures? I wonder, because the next sentence in the text describes other results in these figures properly: “The highest daily UHI impact on mortality risk decreases from 1.05 additional deaths per 100,000 population per day (Budapest, Hungary; Fig. 3a) to 0.61 additional deaths per day (Bucharest, Romania; Fig. 3b), both during heat extremes”.

Apologies for the confusion. Indeed, the first sentence incorrectly referenced figure 3. The “heat/cold days” mentioned here refer to all days in a year warmer/colder than the optimal temperature. This is the same metric as the annual estimates in Fig. 5a,b. To avoid confusion, we’ve removed these two sentences.

c) Line 181: Does figure 3d really show the impact on mortality risk estimation? I have the impression that the word “this” applies rather to the 85+:20-44 population ratio for the city vs. for the standard population.

Sorry for the confusion. Fig. 3d did indeed only show the difference in age structure. “This” referred to both the fact that populations are older in western Europe and subsequently, greater impact on mortality risk estimation is also generally found in these regions.

However, this discussion has now been removed and the figure changed following correction of a bug in the analysis code which brought the local age structure closer to that of the standard population.

I-2) Methodology

My main expertise is in economics, although I have been working with epidemiologists for over 25 years. This may explain why the epidemiological methodology did not seem to me to be detailed enough in the text or in section 4 Methods, subsection 4.2 Epidemiology.

A) Epidemiological methodology issues

It essentially refers to an article of which Pierre Masselot (one of the co-authors of the present work) is also the author: Lines 384-86 “Temperature-mortality exposure-response relationships for five age groups (20-44, 45-64, 65-74, 75-84, and 85+) in each city are obtained from, and explained in more detail in, Masselot et al. [34]”. Yet this publication is neither correctly cited in the bibliography ([34] Masselot, P. et al. Excess mortality attributed to heat and cold in 801 cities in Europe), nor a priori accessible on the net.

Apologies for this. This article was under review and has recently been published. The bibliography has been updated and the details can now be found on the net ([https://doi.org/10.1016/S2542-5196\(23\)00023-2](https://doi.org/10.1016/S2542-5196(23)00023-2)). The associated R code is available on <https://github.com/pierremasselot>.

This led me to three questions of understanding.

a) Lines 72-4: “Mortality risks are explored across urbanisation gradients using high resolution urban climate simulations, population density data, and city level temperature mortality relationships”. The authors use population-specific relationships for each city, which provide a good level of precision. However, in doing so, these relationships already incorporate differences due to UHI in the calculation of mortality risks (see Figure S8), before applying them to temperature variations related to UHI and calculating the frequency weighted UHI impact by temperature. What is the influence of these temperature-mortality relationships on the computations?

Thank you for the comment and question. Just to clarify, Figure S8 was plotted for illustration purpose only, mainly to show why the annual net impact may be minimal or protective when the daily impact is greater during heat extremes. Other analyses in the rest of the study are not based on the approach shown in this figure.

It is true that the city-level exposure-responses already include part of the UHI effect, which is incorporated in the second stage of the epidemiological analysis by estimating the relationships between cities with different average values of this indicator. However, these exposure-responses cannot capture within-city differences in UHI, which we argue is the more important part and is the part evaluated by our analysis through use of UrbClim data. In this context, the results of our analysis provide the correct interpretation even in the presence of between-city baseline differences in the UHI prevalence.

To further clarify, for each city, mortality in our analysis is related to the prevalent environmental temperature. Whether these temperatures are boosted or not due to UHI in the city, the relationships would still be valid. Additionally, city-specific exposure-responses are assumed to identically apply to the whole population in the city, separated by age groups, irrespective of their individual actual exposure or vulnerability, i.e., it is the mean estimate. Therefore in the attribution stage, the point estimate of risk for a given daily mean temperature is identical regardless of where in the city it is observed.

The following sentences are now added to the Epidemiology section under Methods in the manuscript:

“These relationships quantify the mean risk for each city- and age-specific population for each daily mean temperature, irrespective of factors such as UHI that led to the observed temperature. They are assumed to identically apply to the whole population in each age group in each city, irrespective of differences in each individual’s actual exposure or vulnerability, i.e., it is the mean estimate.”

b) Lines 253-54: “The former therefore includes the impact of chronic exposures while the latter only capture acute effects (with lag periods of up to three weeks, see Methods).” A lag of 3 weeks is mentioned, referring to the Methods section, but this section does not give any further details. What are the implications of this “up to three weeks” lag in the calculations, particularly those associated with extreme heat and cold days? Are the impacts on mortality considered to be spread over three weeks? What happens if several consecutive days are classified as extreme? Is there a harvesting effect?

Apologies for missing this in the methods description. In the distributed lag non-linear models used to model the temperature-mortality relationship, we consider lagged effects of each exposure for up to three weeks after the exposure day. The shape of the lagged response is also captured by the statistical model, so for cold exposure, this tends to imply a protracted period of elevated risk, while for heat, the impact is usually a strong peak in risk immediately after exposure. Since we are looking at the cumulative impact of single exposure days, we don’t capture the compounding effect of having several consecutive days of temperature extreme. However, harvesting effect within the three weeks lag period is captured. A note on this has now been added to the methods section and some of the implications added in the discussion section, as appended below.

In the methods section:

“Lagged effects up to three weeks after exposure are considered in the DLNM and summed to represent the short-term cumulative impact of each exposure day.”

In the discussion section:

“Lastly, it should be noted that the impact of cold weather tends to be spread out over a longer period after exposure while that of heat tends to be concentrated on or shortly after the exposure day. Thus, despite similar cumulative impacts on mortality, heat and cold weather have different implications for health care.”

c) Lines 317-21: It is suggested to “capture the true exposure experienced by the population”. In addition to the points rightly raised by the authors, more generally, deaths seem to be associated

with people's place of residence. If people move around the country or abroad during the hottest 2% of days (often in the summer, during holidays), how are deaths counted if they do not occur in their place of residence? What proportion of people are in their main residence during heatwaves?

Deaths are counted by people's residential address, so indeed people being on holiday also affects the true exposure experienced by the population. According to Eurostat, in each year, around 60% of the European population partakes in tourism away from their place of residence for at least one night. This peaks in the third quarter of each year, and can vary a lot by country, with Sweden for instance seeing almost everyone above 15 years old travelling for personal or business reasons away from their place of residence for at least one night during peak season. While a clear indicator of the relevance of this topic, it doesn't translate directly into how each city's population count changes during tourism season, and therefore how many people are truly exposed during heatwaves, given differing timing and duration of trips, as well as inbound tourists. The topic of varying exposure due to population mobility is something we're very interested in and that we're looking into in more detail in separate studies. In the meantime, a comment, as appended below, has now been added in the manuscript noting mobility due to tourism as an additional factor affecting population exposure.

"Additionally, in each year, around 60% of the European population participates in tourism away from their place of residence for at least one night. With peak tourist season in the third quarter of the year when heat extremes also tend to occur, a seasonal factor can further mediate the impact of mobility on population exposure."

Reference: Eurostat data codes: TOUR_DEM_TOTOT, TOUR_DEM_TOQ, DEMO_PJANBROAD

B) Other methodological issues

a) The place of residence differs from the place of work (generally in UHI), which the authors acknowledge. This involves (on average) a higher exposure to high temperature during day w.r.t the place of residence. Is there evidence in the literature of the magnitude of the discrepancies involved?

We are not aware of literature that quantifies the difference in temperature exposure due to difference between work and residential addresses. It is something we are currently investigating in a separate study by looking at the difference in daytime and nighttime populations (Batista e Silva et al. 2020, <https://doi.org/10.1038/s41467-020-18344-5>). Preliminary findings indicate a seasonal cycle and tendency towards greater daytime total population in the city as people travel in to work, though with exceptions (Fig. 1 below). For the case of London, it was found that people tend to travel to warmer parts of the city during the day; the warmer, the greater increase in population (Fig. 2 below). This trend is weaker during later spring and summer months, but picks up again by autumn, when heat extreme events may still be observed. We decided not to include this analysis in the manuscript as it's still ongoing and will be investigated in detail separately.

Fig. 1: Seasonal variation in total nighttime and daytime population in Milan, London, and Stockholm. The nighttime population is based on registered address of the residents and data from online booking platforms. The daytime population is derived based on the locations of the primary daily activities of 16 population groups include residents, employees, students, non-working population, etc. City total is the sum over each city's domain from Batista e Silva et al. (2020).

Fig. 2: Difference between daytime and nighttime population as function of grid monthly average temperature for 2015 in London. Note that the year of reference for daytime and nighttime population data is 2011 (Batista e Silva et al. 2020), so the results should just be

seen as an example of how using daytime or nighttime population changes the estimated population exposure to different temperatures.

b) Are the conditions of health care (speed of intervention, transport to a health care facility) during the health event causing the death the same between UHI and non-UHI? Can this influence mortality between residents in and out of HUI?

Thank you for highlighting this. Yes indeed, access to healthcare and other interventions may differ between urban and rural communities, which can influence health outcomes. We do not consider these differences in our analysis, which focuses on the role of temperature differences due to UHI alone. However, this is now noted in the discussion section.

C) Remarks on economic issues

I agree with the value of statistical life (VSL) chosen to assess mortality impacts, which was carefully calculated by the OECD in 2012 and widely used since then. I am also satisfied with the way it is translated into a value relevant for Europe. Note, however, that some countries recommend that the VSL should be changed according to changes in Gross Net Product per capita instead of Consumer Price Index, but the differences are limited. I have three points to make.

a) First, I am reluctant to the use of the terms “costs” and “benefits” when associated with non-market impacts, here the valuation of mortality. This can be misleading to the reader as these amounts will not actually be gained or lost by the cities concerned, rather they are welfare losses in their population associated with premature mortality and valued economically. I prefer the terminology “economic impacts” or “economic assessments”, which can be negative or positive. I would then be satisfied if this costs/benefits formulation was changed throughout the text. This includes the abstract “UHIs in Europe cost/save €480/€677 in heat-/cold-related mortality”, as well as the title “Mortality costs and savings of urban heat islands in European cities”. For the latter, something like “Economic valuation of acute mortality attributed to urban heat islands in European cities” would be better.

Thank you for the suggestion. We have now changed the title and amended the wording throughout the manuscript. We removed the word “acute” from the suggested title because, even though we don’t capture impacts that may affect long term health, we do consider mortality occurring within 3 weeks after exposure. This is to avoid misunderstanding with acute mortality in the context of heat-related deaths occurring within a day or two after the event as opposed to cold-related deaths occurring over a couple weeks. We specify instead “temperature-related mortality”.

b) Second, in choosing this VSL, the economic computations assume that life expectancy at death is similar to that observed for all causes of death. Since the effects acute mortality are being assessed here, it would be useful to discuss the expected reduction in life expectancy. A difference in life expectancy, and the extreme case of a single year of life lost per death attributable to UHI temperature conditions, would alter the economic computations.

Thank you for the comment and suggestion. We have now added an analysis with years of life lost and value of life year. Discussion on this has now been added in the sections on impacts of UHIs and on economic assessments, also appended below.

In the section on impacts of UHIs (~L206 in the tracked version):

To additionally consider the age dependence of UHIs' impact on mortality, years of life lost (YLL) analyses are included in the Supplementary materials (Fig. S13, Tables S5 and S9). While the overall conclusion of a weakly protective annual net impact for most cities still holds, an annual adverse impact is found for more cities with the YLL approach (29, vs. 18 with the mortality counts-based analysis). As younger age groups are weighted more strongly in YLL analyses, this finding may be reflective of younger populations' greater vulnerability to heat, which is more similar to that of older age groups, compared to their vulnerability to cold, which tends to be lower (Fig. 1a-c).

In the section on economic assessments (~L326 in the tracked version):

Economic assessments by VSL as presented above does not explicitly consider age, therefore life expectancy at time of death is assumed to be comparable to all causes of death. However, as can be observed from age-dependent temperature-mortality relationships (Fig. 1a-c), heat and cold exposure disproportionately affect older populations. The life expectancy at the time of death would therefore likely be shorter than the average for all-cause mortality. An approach to account for this is to consider the YLL through value of life year (VOLY) valuation, which is included in Supplementary table S6. While there is strong correlation between economic impacts as determined through VSL and that through VOLY (Supplementary fig. S17a,b), the magnitude of the impact as assessed through VOLY is only around 14% (median of the multi-city median annual net, heat, and cold impacts) of that assessed through VSL (Supplementary fig. S17c). This is mainly due to differences in valuation approaches which resulted in each VSL (3.91 million 2021-EUR per statistical life [47]) equating to 85 VOLY (46,000 2021-EUR per year [48]). However, there is currently no clear consensus on economic assessments by YLL, and at the higher end of the VOLY estimate (116,000 2021-EUR per year [48]), each VSL equates to around 34 VOLY, resulting in valuations by VOLY-based assessments to be around 35% of that using VSL (Supplementary fig. S17d). Given limitations with the VOLY approach, including ethical concerns and lack of evidence in the assumption of a time-independent VOLY [47], VSL-based assessments are the main focus in this study.

c) Related to the point above, I find that the authors too quickly eliminate the possibility of conducting an analysis based on years of life lost (lines 453-57), and a valuation that would calculate them on the basis of ages of death (at least, age groups) and exposition response functions. This would indeed take into account at least the loss of life expectancy associated with age (even if not with cause). This was done by Khomenko et al (2021) for PM2.5 for example. It would not be necessary to go as far as monetarization (even if some values of life year lost exist for Europe, such as in Desaiques et al., 2011), but at least to make a net balance of years of life gained/lost that would take into account the age distribution.

Thank you for the suggestion. Quantification of years of life lost, as well as monetarisation following Desaiques et al. 2011, have now been added. Discussions are included in the section on impacts of UHI (lines 206-221 in the tracked changes version) and the section on economic assessment (lines 326-344 in the tracked version), also included in the reply to the previous comment.

II) Use of statistics

In the box-and-whisker plots (Figures 3a et b, as well as S3, S7, S10 S14 and S16), please specify what the values represent (certainly the median for the centre line, quartiles 1 and 3 for the box, 1.5 the interquartile range (IQR) Q3-Q1 for the whiskers (with the respective boundaries the minimum and maximum), and outliers for dots outside the IQR).

Thank you for the suggestion. Now added.

III) List of suggested improvements

Although this work is already very comprehensive, I think that the benefits of implementing some of the 4 directions I present below are worth considering.

a) The authors compare their results with those for mortality associated with PM2.5 exposure. The latter is, in general, proportional to population density in cities, and thus acts in synergy with the effects of temperature for UHI, both in summer and winter. However, another pollution indicator appears to be related to heat wave mortality: ground-level ozone (e.g. Reid et al, 2012). In this case, the relationship with UHI would be rather antagonistic in summer or even on hot spring or autumn days, since in highly polluted areas, and under certain insolation conditions, high NOx concentrations can destroy part of the ozone. Consequently, ozone levels are likely to be higher in peri-urban and rural areas downwind of UHI. To what extent can this affect the results?

Thank you for this comment. We indeed agree with the reviewer that ozone is generally lower in central urban areas and higher in the suburbs and surroundings, since ozone formation takes time and the reduction in VOC emission in cities means that ozone formation now requires some of the biogenic emissions coming from vegetation around cities (e.g., see analysis by one co-author in Llaguno-Munitxa and Bou-Zeid 2020, <https://iopscience.iop.org/article/10.1088/1748-9326/ab8d7e>). However, to deduce whether ozone concentrations in cities will in fact anticorrelate or correlate with temperatures during extreme heat events, one has to also account for multiple confounding processes that affect the ozone concentrations here including lower typical wind speed during extreme heat (increases ozone), higher insolation (increases ozone), and altered vegetation activity (might reduce ozone if plants shutoff during extreme heat, or increase it if they transpire more to control their temperatures). As the mentioned Reid et al. 2012 paper summarised, there is no consensus on the treatment of ozone in temperature-related health impact assessment studies.

In the manuscript we have now added a comparison of annual heat and ozone mortality, and in fact we find a positive correlation as with PM2.5, whereby cities with greater heat mortality risk also tend to experience greater ozone-related mortality risk. This is, however, on an annual scale as daily air pollution-related mortality data is not available. Additionally,

ozone-related risk is quantified against thresholds set by WHO guidelines as opposed to the rural values as is the case for UHI. This does not contradict the fact that during periods of extreme heat, ozone concentrations might still be more elevated in rural terrain; therefore, when correlation in time are considered, both ozone and heat increase mortality in urban and rural areas during heat waves. A discussion on this is included in the manuscript along with the additional ozone analysis. (paragraph starting on line 298 in the manuscript with tracked changes)

b) Men and women have a different reactivity to temperature stress. This is linked to physiological factors: muscle mass, which is higher on average in men, generates heat, whereas fat mass, which is higher on average in women, does not. It is also linked to different lifestyles: fewer women are active, implying differences in exposure at work or when travelling to work. The authors seem to have the annual mortality by age and sex. If other data are available by sex too, would it be possible to perform the calculations in step 4.2 separately by sex?

Unfortunately, this is not possible as it would require daily mortality counts separated by sex, on top of age, for all cities used in developing the exposure-response relationships, as well as sex-specific observations to be used in the meta-analysis, which are not fully available. We agree that some differences in risk may be present and would be worth exploring, though it's not the focus of this study since we are focusing on overall city impacts and the risk relationships here are already representative of the sex make-up of each city.

c) The authors discuss the consequences of climate change on their analyses. Lines 276-81, it is written: "Additionally, climate change is expected to lead to increases in the proportion of warm days, and therefore heat-related risks, relative to cold days and cold-related risks in future [47]. Given the weak annual net protective effect of UHI for most cities examined and the greater impact per day for heat, temperature increases in future can quickly shift the balance and lead to an annual net adverse impact". Why not make these estimates? Gasparrini et al. (2017) did so on the basis of different representative concentration pathways (RCPs). Why not also apply it to UHI under climate change scenarios? It seems to me that the authors can easily obtain all the data they need. And this will eventually allow to determine, for each city, when the adverse effects of high temperature UHI will outweigh the protective effects of low temperature UHI.

Thank you for the suggestion. While in principle it may be possible to provide a first order estimate of climate change impacts, for instance by applying a uniform shift to the temperature distribution, we feel a more sophisticated approach is necessary especially in the context of UHI. An important distinction between our study and for instance Gasparrini et al. (2017) is that UHI is not a static quantity. The temperature difference between urban and rural areas will be strongly affected by urban growth and decline in future, and it varies significantly in time especially during heat and cold waves. Even without infrastructural changes, the response of urban temperatures to climate change may not be the same as the response in rural areas (e.g. Zhang et al. 2023, <https://iopscience.iop.org/article/10.1088/1748-9326/acbc90>). Climate change projections are based on coarse-grid global climate models, which often have minimal representation of urban areas. Therefore, to examine UHI, urban downscaling would generally be required, often in time slices. In the case of our study setup, model simulations using UrbClim would need to be performed under future conditions, which would need to be properly constrained. On the epidemiology side, future population growth and aging will also alter

the quantification of impact, even if vulnerability of each age group is assumed to remain the same (e.g. no advances in healthcare). Given these considerations, we feel that appending a first order estimate by assuming uniform shifts in temperature and UHI may be more misleading than helpful. Instead, a more complete consideration of UHI impact in future under climate change would be required, which goes beyond the scope of this study.

d) In Line 337, the authors mention in future work a more detailed study of vulnerabilities. Why would it not be possible to make a first attempt to construct indicators of deprivation and to match them at grid level? Several indexes are possible.

- The European Deprivation Index within urban areas, based on EU – SILC (European Union – Statistics on Income and Living Conditions) can be exploited, or build from aggregated census data and tax incomes of the population by year.

- A simpler version, originally constructed in France (Windenberger et al., 2012), is based on 4 variables: the median household income per consumption unit, the percentage of high-school graduates in the population aged 15 years and older, the percentage of blue-collar workers in the active population aged 15 to 64, and the unemployment rate in the active population aged 15 to 64.

- The Townsend deprivation index (Townsend, 1987), originally constructed in England and Wales, is also based on four variables: the rate of households without a car, the rate of non-owner households, the rate of households with more than one person per room, and the rate of unemployment in the active population.

Thank you for the suggestions, especially of the deprivation indices. Due to the limited availability of granular deprivation and associated census data at intra-urban resolution, we are not able to perform this analysis for many cities. However, we were able to obtain at high spatial resolution the Index of Multiple Deprivation for England, and analyses for London and Leeds as example cities are now added in the supplementary materials (Fig. S18), with a discussion added in the Discussion section in the main text (line 417-431 in the tracked version), also appended below.

Examination with the 2019 English Index of Multiple Deprivation [64] for example cities of London and Leeds in the current study also shows statistically significant, though moderate (Spearman's correlations of -0.4 and -0.6, respectively), correlations between socioeconomic deprivation and UHI-induced mortality risk during heat extreme days (Supplementary fig. S18). More socioeconomically deprived neighbourhoods tend to experience greater UHI impact on average, though similar UHI impacts are also observed in neighbourhoods with significantly different deprivation levels (Supplementary fig. S18b,d). In addition to indicating a biased exposure for more socioeconomically deprived neighbourhoods, this also highlights a potential overlap between neighbourhoods with higher UHI and those with greater vulnerability. Populations in more socioeconomically deprived neighbourhoods may be more vulnerable due to factors such as lower income, poorer health, inferior living environment, and greater barrier to housing and services, all of which are included in the 2019 English Index of Multiple Deprivation [64]. The greater vulnerability on top of higher exposure may therefore further compound the overall impact of UHI.

IV) References

- Desaigues, B., Ami, D., Bartczak, A., Braun-Kohlová, M., Chilton, S., Czajkowski, M., Farreras, V. et al. (2011) Economic valuation of air pollution mortality: A 9-country contingent valuation survey of value of a life year (VOLY). *Ecological Indicators* 11, 902–910
- Gasparrini, A., Guo, Y., Sera, F., Vicedo-Cabrera, A. M., Huber, V., Tong, S., ... & Armstrong, B. (2017). Projections of temperature-related excess mortality under climate change scenarios. *The Lancet Planetary Health*, 1(9), e360-e367
- Guillaume, E., Pornet, C., Dejardin, O., Launay, L., Lillini, R., Vercelli, M., Marí-Dell'Olmo, M., et al. 2016. Development of a Cross-Cultural Deprivation Index in Five European Countries. *Journal of Epidemiology and Community Health* 70(5), 493–99.
- Khomenko, S., Cirach, M., Pereira-Barboza, E., Mueller, N., Barrera-Gómez, J., Rojas-Rueda, D., ... & Nieuwenhuijsen, M. (2021). Premature mortality due to air pollution in European cities: a health impact assessment. *The Lancet Planetary Health*, 5(3), e121-e134.
- Reid, C. E., Snowden, J. M., Kontgis, C., & Tager, I. B. (2012). The role of ambient ozone in epidemiologic studies of heat-related mortality. *Environmental health perspectives*, 120(12), 1627-1630.
- Townsend, P. (1987). Deprivation. *Journal of Social Policy*, 16(2), 125-146.
- Windenberger, F., Rican, S., Jougl, E., & Rey, G. (2012). Spatiotemporal association between deprivation and mortality: trends in France during the nineties. *The European Journal of Public Health*, 22(3), 347-353.

Reviewer #3 (Remarks to the Author):

The article is innovative and brings a significant contribution to the research in the field of urban heat island and mortality under heat and cold conditions. The study area is wide-ranging, and all the most advanced methodologies have been applied to investigate the relationship between mortality and temperature. The use of UrbClim makes the analyses (partially) integrated with the characteristics of the built environment and the Köppen-Geiger climate classification ensures that the classifications used are validated and integrated with the current literature. The graphs are accurate and understandable. Finally, the inclusion of micro-particle pollution is an added value, although it would have been interesting to see the association between O₃ and PM₁₀, 2.5 and heat- and cold- related deaths.

Review of the paper:

“Mortality costs and savings of urban heat islands in European cities”

First of all, I would like to congratulate the authors for the excellent work. The conducted analyses are in line with the up-to-date scientific literature and the discussion of the results is very rich and interesting. This manuscript brings a significant contribution to the research in the field of urban heat island and mortality under heat and cold conditions within EU urban areas. The most advanced methodologies have been applied to investigate the relationship between mortality and temperature and the use of UrbClim makes the analyses (partially) integrated with the characteristics of the built environment. In addition, the Köppen-Geiger climate classification

ensures that the classifications used are validated and integrated with the current literature. The inclusion of micro-particle pollution is an added value, although it would have been interesting to see the association between O₃ (in summer) and PM₁₀, 2.5 (in winter). Notwithstanding, contents and technical details are of excellent quality.

Thank you for the positive feedback and suggestions. An additional analysis for ozone-related mortality and how it relates to temperature-related mortality has now been added in the discussion about comparison to air pollution (paragraph starting on line 298 in the manuscript with tracked changes). Though not the focus of the main text, associations between O₃- and PM_{2.5}-related mortality (annual total estimates) have also been added in the supplementary materials (Fig. S16f).

As minor comment, I would like to raise two points:

1] A better clarification of the meaning of “*exposure*”, “*vulnerability*”, and “*risk*” in the context of this research would not be a useless explanation. Especially in the climate change field, many of these concepts/words blur getting sometimes meanings sensibly different according to the reader’s background. For this reason, I believe that a clear definition of the used terms would be of help to close the gaps between the environmental epidemiological and other interdisciplinary studies (see IPCC Glossary, IPCC 2022).

Thank you for the suggestion. A new paragraph has now been added at the end of the introduction section to define these words.

2] A clear identification of social inequalities in the nexus between extreme temperatures and mortality would have provided a clearer understanding of the risk identification across the different categories of the population (most educated vs least educated, most isolated vs least isolated, etc.), but we are aware that the availability of data at the urban level is not the same in all cities. Based on the results produced by the study (e.g., *Table S2 “Mortality associated with UHI, per 100,000 urban adult population”*), it might be interesting to further investigate these aspects in some cities where similar data are available.

Other punctual recommendations in the text:

On Results:

Line 95: insert a “,” after “e.g.”

A comma has now been added to all instances of e.g. and i.e. in the manuscript.

Line 126: it could be interesting to see some examples “*of cities which reach around the 20% of enhancement*”.

On Discussion:

- The use of the term “*mitigation*” here can lead to misunderstandings. I would suggest using other terms in line with the IPCC and UNFCCC languages (i.e., “*adaptation*”) (IPCC 2022), when appropriate.

Reference in the text:

Line 267: “*this indicates the need for mitigation strategies*”

Line 299: “such analyses can then allow a more even comparison to the costs of various mitigation plans”

Thank you for pointing this out. We meant mitigation in the sense of reducing the amount of urban heat, instead of adapting to existing urban heat, though we agree that the word mitigation may be misunderstood as “mitigation of climate change,” in the way of carbon budgeting. Given the importance of UHI mitigation strategies that prevent the build up of urban heat in the first place (e.g. urban greening), as opposed to adaptation strategies that try to cope with the heat (e.g. air conditioning), we’ve kept the word “mitigation” in these two examples listed above, but now noted specifically that we’re referring to “UHI mitigation.” On the other hand, we’ve removed “climate mitigation” in the abstract.

- Spatial heterogeneity could be influenced by outdoor (e.g., direction of the windows front, building age, roofing construction, number of floors) as well as indoor (e.g., thermal isolation, walls and ceiling, fans, double-glazed windows) environmental characteristics. Consider whether to quote them in the text.

Thank you for the suggestion. We have now expanded this point in the manuscript with more examples, as appended below.

Additional spatial heterogeneity in other vulnerability factors, such as properties controlling the temperature of the housing stock associated with both structural features (e.g., insulation, number of floors) and internal environmental regulation (e.g., ventilation, air conditioning), may compound the impact associated with outdoor UHI.

On Methods:

- No citations to DLNM model in the text (section 4.2).

Thank you for pointing this out. Now added.

- No citations to MMT in the text (section 4.4).

MMT is a measure commonly used in temperature-related health assessments to indicate the optimal temperature where the associated risk is at its minimal. We are not able to identify the reference where this measure was first introduced as no reference is given in any of the published literature we found where MMT is used. However, a definition is given in the manuscript so hopefully its meaning is clear.

Line 397: about “UrbClim domain average temperature from the 2008 to 2017 period”. Why did you consider only 9 years? Is the available timeframe? For the World Meteorological Organization, 30-year reference period reflects changing climate.

Yes, 2008 to 2017 is the full range of UrbClim simulations available. Given the biases that can exist between different temperature datasets, and given that UrbClim provides temperature across the entire urban domain at high spatial resolution, we felt it more important to use the UrbClim temperatures rather than a longer time series from another dataset.

Line 313-316: In relation to “A previous study has found, for instance, that within Parisian neighbourhoods experiencing the greatest heat exposure, those which are the most socioeconomically deprived have a twofold higher mortality risk compared to the most affluent [53]”.

To provide a clearer overview of the state-of-the-art in relation to social inequalities - extreme temperatures – health, I suggest referring to other cities when considering the relation between temperature and mortality and different health outcomes socio-economic variables. One option may be to quote studies over cities with a higher mortality ranking associated with UHI (Table S2 in Supplementary Material) (e.g., Turin, Madrid, Bologna, Thessaloniki, etc.).

Thank you for the suggestion. Unfortunately, we were unable to find publications in the literature looking at correlations between socioeconomic status and temperature exposure for these cities. High resolution data of socioeconomic status for these cities is also not readily available for us to perform the analysis ourselves. However, we were able to obtain the Index of Multiple Deprivation for England at high spatial resolution, and analyses for London and Leeds as example cities have now been included in the manuscript (line 417-431 in the tracked version). The associated text in the discussion is appended below.

Examination with the 2019 English Index of Multiple Deprivation [64] for example cities of London and Leeds in the current study also shows statistically significant, though moderate (Spearman's correlations of -0.4 and -0.6, respectively), correlations between socioeconomic deprivation and UHI-induced mortality risk during heat extreme days (Supplementary fig. S18). More socioeconomically deprived neighbourhoods tend to experience greater UHI impact on average, though similar UHI impacts are also observed in neighbourhoods with significantly different deprivation levels (Supplementary fig. S18b,d). In addition to indicating a biased exposure for more socioeconomically deprived neighbourhoods, this also highlights a potential overlap between neighbourhoods with higher UHI and those with greater vulnerability. Populations in more socioeconomically deprived neighbourhoods may be more vulnerable due to factors such as lower income, poorer health, inferior living environment, and greater barrier to housing and services, all of which are included in the 2019 English Index of Multiple Deprivation [64]. The greater vulnerability on top of higher exposure may therefore further compound the overall impact of UHI.

REVIEWER COMMENTS

Reviewer #2 (Remarks to the Author):

This revised work studies the impacts of temperature changes due to urban heat island (UHI) on mortality, across 85 European cities belonging to 5 background climatic conditions. It accounts for intra-urban variations in temperature and population distribution and compute the health and economic impacts over the entire annual cycle, as well as the net effect.

The conclusions are original in that they provide results on the net balance of UHI benefits over the whole year, for numerous cities covering a variety of climatic conditions. They can help decision-makers understand the impact of UHIs on (short-term) mortality and consider policies to reduce them at city level. The conclusions and interpretation of the data are robust, valid and reliable, with many additional analyses supporting the main results, some of them responding to my comments on the original version (see below). The references seem adequate.

I explain below how the authors have taken my comments into account in the revised version, and end with three remarks.

* The revised version adequately addresses my three comments on the reporting of the results.

* Regarding methodology, it now provides the appropriate (and up-to-date) reference to the main article that presents the epidemiological methodology used (Masselot et al., 2023).

* Regarding epidemiological methodological issues, the authors have correctly addressed my comments concerning the influence of city-level temperature-mortality relationships on calculations of the impact of UHIs within the city, how lag periods between exposure and mortality effects are taken into account, and the potential influence of mobility due to tourism on the population exposure during the hottest days.

* Regarding other methodological issues, I thank the authors for sharing, in their rebuttal, preliminary results on seasonal variation in the total nighttime and daytime population in response to my comments on potential differences between place of residence and place of work. They also include in the discussion section of the article my remark on potential differences in healthcare conditions between UHIs and non-UHIs.

* With regard to economic issues, I am satisfied with the replacement of the terms "costs" and "benefits" by the terms "economic impacts", "economic assessments" or "economic evaluation" throughout the article (including the title). Similarly, I am satisfied with the additional analysis in terms of years of life lost / life expectancy which I requested and which is now included in the main text as well as in the Supplementary Appendix.

* Regarding the use of statistics, the description of box and whisker diagrams has been clarified in Figures 3 and 5, as well as in the Appendix.

* The authors have taken into account the four additional directions I suggested in my initial report, providing different types of response.

- With regard to ozone, the correlation between ozone-related deaths and heat-related deaths is presented in Figure 5e at city level, the comparison with the costs of heat-related deaths is presented in Figure 5d, and a few sentences explain the results in the main text.

- As regards sex-specific calculations, the authors explain that they are not possible because the daily mortality counts and sex-specific observations to be used in the meta-analysis are not fully available.

- As for the impact of climate change on their analyses, the authors replied that it would require either a very detailed analysis at the UHI level, thanks to urban downscaling which is outside the scope of their study, or a potentially spurious assumption of uniform temperature changes which could be more misleading than helpful.

- Regarding my suggestion to make a first attempt to match indicators of deprivation and temperature-related health effects at grid level, the authors have made an initial exploration. They focused on UHI-induced mortality among 65–74-year-old on the warmest 2% days in two cities, London and Leeds, based on the availability of the English Index of Multiple Deprivation. The result was a negative and statistically significant correlation between UHI-induced mortality and socioeconomic deprivation in both cities (see Table S18).

* Remarks

- As indicated above, the exploration of a possible negative relationship is currently limited to the 65-74 age group, which should be specified in the main text.

- Furthermore, given that the mortality RR for this age group appears to be more sensitive to cold than to high temperatures in London (see Figure 1b and Figure S8e), and probably also in Leeds, it would be interesting to also calculate the correlation between UHI-induced mortality and socio-economic deprivation during the coldest 2% days. Indeed, Tables S3, S5, S7 and S9 show that the effect per 100,000 urban adult population for these days is comparable for London, but three times greater for Leeds. Moreover, if we compare cold days with heat days, the same four tables indicate that the effects are 6 times greater for London and around 16 times greater for Leeds. Therefore, it would be interesting to check whether the negative correlations obtained during the 2% warmest days are also obtained during the 2% coldest days, and/or during heat and cold days.

- Finally, it should be noted that the economic impacts provided in Tables S4, S6, S8 and S10 are calculated per adult and not per 100,000 adults, as incorrectly stated in the titles.

Olivier Chanel
Aix Marseille Univ, CNRS, AMSE, Marseille, France

Thank you very much for the feedback. Please find below in blue responses to individual remarks. Additionally, we have updated the exposure-response relationships used in this study to be consistent with the published version associated with Masselot et al. 2023 and added uncertainty quantifications of the mortality estimates using the Monte Carlo simulations therein. This resulted in some minor changes in the estimates and associated discussions, but the overall findings remain unchanged. Please find attached documents with tracked changes for both the manuscript and the supplement.

Reviewer #2 (Remarks to the Author):

This revised work studies the impacts of temperature changes due to urban heat island (UHI) on mortality, across 85 European cities belonging to 5 background climatic conditions. It accounts for intra-urban variations in temperature and population distribution and compute the health and economic impacts over the entire annual cycle, as well as the net effect.

The conclusions are original in that they provide results on the net balance of UHI benefits over the whole year, for numerous cities covering a variety of climatic conditions. They can help decision-makers understand the impact of UHIs on (short-term) mortality and consider policies to reduce them at city level. The conclusions and interpretation of the data are robust, valid and reliable, with many additional analyses supporting the main results, some of them responding to my comments on the original version (see below). The references seem adequate.

I explain below how the authors have taken my comments into account in the revised version, and end with three remarks.

* The revised version adequately addresses my three comments on the reporting of the results.

* Regarding methodology, it now provides the appropriate (and up-to-date) reference to the main article that presents the epidemiological methodology used (Masselot et al., 2023).

* Regarding epidemiological methodological issues, the authors have correctly addressed my comments concerning the influence of city-level temperature-mortality relationships on calculations of the impact of UHIs within the city, how lag periods between exposure and mortality effects are taken into account, and the potential influence of mobility due to tourism on the population exposure during the hottest days.

* Regarding other methodological issues, I thank the authors for sharing, in their rebuttal, preliminary results on seasonal variation in the total nighttime and daytime population in response to my comments on potential differences between place of residence and place of work. They also include in the discussion section of the article my remark on potential differences in healthcare conditions between UHIs and non-UHIs.

* With regard to economic issues, I am satisfied with the replacement of the terms "costs" and "benefits" by the terms "economic impacts", "economic assessments" or "economic evaluation" throughout the article (including the title). Similarly, I am satisfied with the additional analysis in terms of years of life lost / life expectancy which I requested and which is now included in the main text as well as in the Supplementary Appendix.

* Regarding the use of statistics, the description of box and whisker diagrams has been clarified in Figures 3 and 5, as well as in the Appendix.

* The authors have taken into account the four additional directions I suggested in my initial report, providing different types of response.

- With regard to ozone, the correlation between ozone-related deaths and heat-related deaths is presented in Figure 5e at city level, the comparison with the costs of heat-related deaths is presented in Figure 5d, and a few sentences explain the results in the main text.

- As regards sex-specific calculations, the authors explain that they are not possible because the daily mortality counts and sex-specific observations to be used in the meta-analysis are not fully available.

- As for the impact of climate change on their analyses, the authors replied that it would require either a very detailed analysis at the UHI level, thanks to urban downscaling which is outside the scope of their study, or a potentially spurious assumption of uniform temperature changes which could be more misleading than helpful.

- Regarding my suggestion to make a first attempt to match indicators of deprivation and temperature-related health effects at grid level, the authors have made an initial exploration. They focused on UHI-induced mortality among 65–74-year-old on the warmest 2% days in two cities, London and Leeds, based on the availability of the English Index of Multiple Deprivation. The result was a negative and statistically significant correlation between UHI-induced mortality and socioeconomic deprivation in both cities (see Table S18).

* Remarks

- As indicated above, the exploration of a possible negative relationship is currently limited to the 65-74 age group, which should be specified in the main text.

Thank you for pointing this out. The 65-74 age group was chosen as a representative example. The relationship is near identical for all other age groups (please see figures below) because the exposure-response curves follow similar monotonic shapes for all age groups in each city. We also do not capture potential differences in spatial distribution of different aged populations. The rank relationship between deprivation and temperature-related mortality is therefore essentially a relationship between deprivation and temperature. To avoid confusion, we have now changed the plot in the supplement to show the relationship for the entire age weighted adult population.

Figure 1: Correlation between socioeconomic deprivation and UHI's impact on mortality risk for (a-e) London and (f-j) Leeds in the UK. UHI impact is shown as the difference in mortality risk compared to the rural average for the (a,f) 20-44, (b,g) 45-64, (c,h) 65-74, (d, i) 75-84, and (e,j) 85+ age groups during the warmest 2% days in 2015-2017. The deprivation index ranks 32,844 LSOA areas in England from most deprived (low IMD rank) to least deprived (high IMD rank).

Figure 2: As figure 1 but for the coldest 2% days.

- Furthermore, given that the mortality RR for this age group appears to be more sensitive to cold than to high temperatures in London (see Figure 1b and Figure S8e), and probably also in Leeds, it would be interesting to also calculate the correlation between UHI-induced mortality and socio-economic deprivation during the coldest 2% days. Indeed, Tables S3, S5, S7 and S9 show that the effect per 100,000 urban adult population for these days is comparable for London, but three times greater for Leeds. Moreover, if we compare cold days with heat days, the same four tables indicate that the effects are 6 times greater for London and around 16 times greater for Leeds. Therefore, it would be interesting to check whether the negative correlations obtained during the 2% warmest days are also obtained during the 2% coldest days, and/or during heat and cold days.

Thank you for the suggestion. Plots for the coldest 2% days have now been added in Supplementary figure S18. We see a positive correlation for cold temperatures, whereby more deprived neighbourhoods tend to experience greater protective effects during cold extreme days. This is consistent with the expectation that the magnitude of UHI

impacts, whether protective or adverse, is correlated with socioeconomic deprivation. This is now also noted in the discussion in the manuscript.

- Finally, it should be noted that the economic impacts provided in Tables S4, S6, S8 and S10 are calculated per adult and not per 100,000 adults, as incorrectly stated in the titles.

Thank you for spotting this. This has now been corrected.

REVIEWERS' COMMENTS

Reviewer #2 (Remarks to the Author):

The second revised version adequately addresses my three comments on the reporting of the results.

Olivier Chanel, Aix Marseille Univ, CNRS, AMSE, Marseille, France

REVIEWERS' COMMENTS

Reviewer #2 (Remarks to the Author):

The second revised version adequately addresses my three comments on the reporting of the results.

Olivier Chanel, Aix Marseille Univ, CNRS, AMSE, Marseille, France

Thank you very much for the valuable and insightful feedback throughout the review process.